# Thalamo-cortical axons regulate the radial dispersion of neocortical GABAergic interneurons

Sabrina Zechel[1], Yasushi Nakagawa[2], Carlos F Ibáñez[1,3,4]*

[1]Department of Neuroscience, Karolinska Institute, Stockholm, Sweden; [2]Department of Neuroscience, University of Minnesota Medical School, Minneapolis, United States; [3]Department of Physiology, National University of Singapore, Singapore, Singapore; [4]Life Sciences Institute, National University of Singapore, Singapore, Singapore

**Abstract** Neocortical GABAergic interneuron migration and thalamo-cortical axon (TCA) pathfinding follow similar trajectories and timing, suggesting they may be interdependent. The mechanisms that regulate the radial dispersion of neocortical interneurons are incompletely understood. Here we report that disruption of TCA innervation, or TCA-derived glutamate, affected the laminar distribution of GABAergic interneurons in mouse neocortex, resulting in abnormal accumulation in deep layers of interneurons that failed to switch from tangential to radial orientation. Expression of the KCC2 cotransporter was elevated in interneurons of denervated cortex, and KCC2 deletion restored normal interneuron lamination in the absence of TCAs. Disruption of interneuron NMDA receptors or pharmacological inhibition of calpain also led to increased KCC2 expression and defective radial dispersion of interneurons. Thus, although TCAs are not required to guide the tangential migration of GABAergic interneurons, they provide crucial signals that restrict interneuron KCC2 levels, allowing coordinated neocortical invasion of TCAs and interneurons.

*For correspondence: carlos. ibanez@ki.se

**Competing interests:** The authors declare that no competing interests exist.

## Introduction

Interneurons of the mammalian neocortex are generated in transient neurogenic structures of the embryonic ventral forebrain, including the lateral, medial, and caudal ganglionic eminences (LGE, MGE, and CGE, respectively), the preoptic area (POA) and the septum (*Bartolini et al., 2013*; *Wonders and Anderson, 2006*). As the developing brain expands, interneurons migrate tangentially to the overlying neocortex over a period of several days in the mouse (*Corbin et al., 2001*). They enter the cortex through a deep path following the subplate and intermediate zone (IZ) and a superficial path in the marginal zone (MZ) (*Marín, 2013*; *Marín and Rubenstein, 2001*; *Wichterle et al., 2001*). The MGE contributes 50–60% of all cortical interneurons, including the majority of parvalbumin and somatostatin-expressing neurons of the neocortex (*Gelman and Marín, 2010*). MGE-derived interneurons develop from precursors that express the transcription factor Nkx2.1 (*Xu et al., 2004*) and are later distinguished by the expression of the LIM/homeobox gene *Lhx6* (*Lavdas et al., 1999*). Several molecular signals have been identified that regulate the tangential migration and dispersion of interneurons towards and within the neocortex, including Neuregulin-1 (NRG1) (*Flames et al., 2004*), hepatocyte growth factor (HGF) (*Powell et al., 2001*), glial cell line-derived neurotrophic factor (GDNF) (*Canty et al., 2009*; *Pozas and Ibáñez, 2005*) and the chemokine Cxcl12 (*López-Bendito et al., 2008*). After their tangential dispersion through the neocortex, interneurons switch their mode of migration from tangential to radial and invade the cortical plate. MGE-derived

interneurons migrating through the IZ move dorsally to occupy positions in different cortical layers; early-born interneurons in layers V and VI, later-born in layers II-IV (*Bartolini et al., 2013*). Despite significant progress in the identification of signals controlling tangential migration of cortical interneurons, the mechanisms that regulate their radial dispersion and laminar distribution are less well understood.

Thalamo-cortical axons (TCAs) make ipsilateral connections between distinct thalamic nuclei and cortical areas, thereby relying sensory information to the neocortex. The development of the thalamo-cortical projection has been widely used as a model system for the study of mechanisms controlling circuit wiring in the mammalian brain (*Garel and López-Bendito, 2014*; *Lopez-Bendito and Molnár, 2003*). There are several intriguing parallels between TCA pathfinding and GABAergic interneuron migration to the neocortex. After crossing the internal capsule —at about embryonic day 13 (E13) in the mouse— TCAs advance through the subpallium following a trajectory that overlaps with that used by migrating GABAergic interneurons exiting the MGE. At around E14, TCAs cross the pallial-subpallial boundary and, similar to GABAergic interneurons, enter the neocortex through the IZ, arriving at the appropriate cortical regions by E16. TCAs then wait in the IZ/subplate for 1 to 2 days before branching, invading the cortical plate and forming synapses at the appropriate layers. A similar waiting period has been observed for GABAergic interneurons entering through the IZ prior to their switching from tangential to radial migration and cortical invasion by E18 (*López-Bendito et al., 2008*). These parallels suggest that GABAergic interneuron migration and TCA pathfinding may be interdependent and/or share common signals.

In this study, we tested the hypothesis that TCAs may provide guidance to MGE-derived GABAergic interneurons for their tangential migration to the neocortex and subsequent radial dispersion and cortical invasion. For this purpose, we studied interneuron distribution and migration in the neocortex of the *Gbx2* mutant mouse, which lacks TCAs as a consequence of abnormal thalamic development (*Hevner et al., 2002*; *Wassarman et al., 1997*). We found that MGE-derived interneurons reached the neocortex in normal numbers in mutant mice lacking Gbx2 either globally or specifically in the thalamus. However, in the absence of TCAs, or TCA-derived glutamate, a significant proportion of interneurons failed to invade the cortex and accumulated in deep cortical layers. Ablation of the KCC2 co-transporter (also known as Slc12a5) rescued this phenotype, indicating that TCAs control radial dispersion of interneurons by supplying signals, such as glutamate, that restrain interneuron KCC2 levels, allowing the normal laminar distribution of neocortical interneurons.

## Results

### Abnormal laminar distribution of GABAergic interneurons in neocortex lacking TCAs

In order to investigate the role of TCAs in tangential migration and radial dispersion of MGE-derived interneurons, we took advantage of the *Gbx2* mutant mouse, which lacks TCAs as a consequence of abnormal thalamic development (*Hevner et al., 2002*; *Wassarman et al., 1997*) (*Figure 1—figure supplement 1*). As mice lacking Gbx2 die shortly after birth, our initial studies were focused on newborn animals. Despite the lack of TCAs, the laminar organization of the *Gbx2* mutant cortex was normal at birth, as assessed by immunostaining for several layer-specific markers (*Figure 1—figure supplement 2*). Likewise, the area specification of prospective visual and somatosensory cortices in newborn *Gbx2* mutants was comparable to that of wild type littermates (*Figure 1—figure supplement 3*). We visualized MGE-derived GABAergic interneurons using the Lhx6-GFP reporter (*Grigoriou et al., 1998*; *Lavdas et al., 1999*) and quantitatively assessed their distribution across three equally-sized sectors encompassing upper, middle and lower layers of the newborn cortex, respectively (*Figure 1—figure supplement 4*). The upper sector included layers II to V, demarcated by the expression of Cux1 (layers II-IV) and CTIP2 (layer V). The middle sector consisted mainly of layer VIa and encompassed the majority of the Tbr1$^+$ territory between the CTIP2 and CTGF markers. The lower sector included layer VIb (or subplate), expressing CTGF, and the underlying intermediate zone (IZ). Across all three primary cortices examined (i.e. prospective M1, S1 and V1), the neocortex of newborn *Gbx2* knock-out mice showed reduced numbers of GABAergic interneurons in the upper layers of the cortical plate compared to wild type littermates (*Figure 1A and B*). The number of GABAergic interneurons in the middle sector was comparable in mutant and wild type cortices (*Figure 1C*). On the

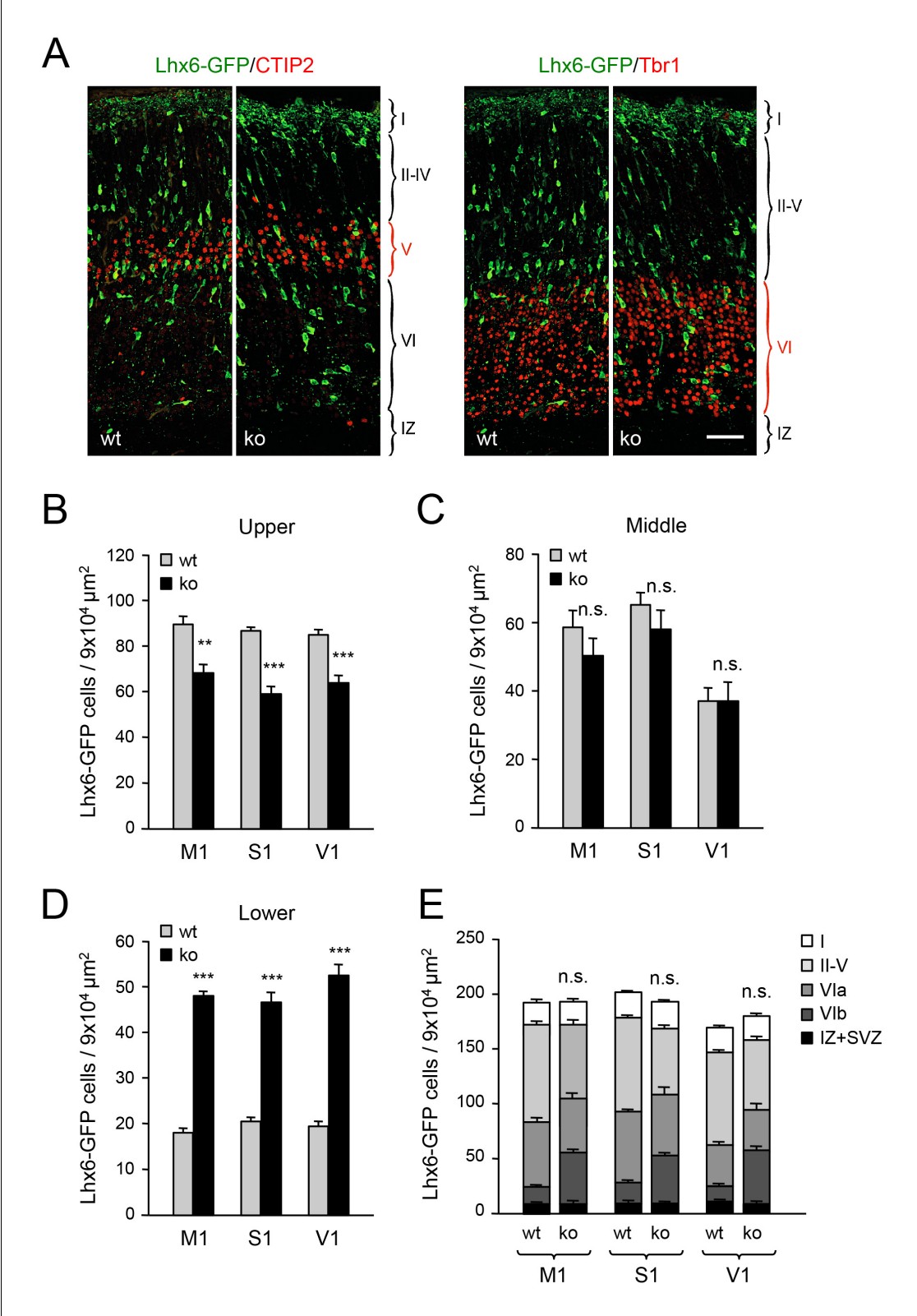

**Figure 1.** Abnormal laminar distribution of cortical GABAergic interneurons in *Gbx2* knock-out mice. (**A**) Lhx6-GFP+ interneurons (green) combined with immunostaining (red) for CTIP2 (layer V marker, left panel) or Tbr1 (layer VI marker, right panel) in prospective somatosensory cortex of newborn wild type (wt) and *Gbx2* knock-out (ko) mice. Cortical layers are indicated. IZ, intermediate zone. Scale bar, 50 μm. (**B–D**) Quantification of Lhx6-GFP+ interneurons in upper (**B**), middle (**C**) and lower (**D**) layers of prospective primary motor (M1), somatosensory (S1), and visual (V1) cortices of newborn

*Figure 1 continued on next page*

*Figure 1 continued*

wild type (wt) and *Gbx2* knock-out (ko) mice. Results are expressed as average ± SEM (***p<0.0001; **p<0.0005; ns, not significant, N = 5 mice in each group). (E) Combined quantification of Lhx6-GFP$^+$ interneurons in all layers of prospective M1, S1 and V1 neocortex of newborn wild type (wt) and *Gbx2* knock-out (ko) mice. Layer VI was subdivided into VIa (Tbr1$^+$) and VIb (subplate, Tbr1$^+$ and CTGF$^+$). No significant difference in the number of GABAergic interneurons was found in the marginal zone (layer I) of the *Gbx2* mutants. Results are expressed as average ± SEM (ns, not significant, N = 5 mice in each group).

The following figure supplements are available for figure 1:

**Figure supplement 1.** Loss of TCAs in *Gbx2* knock-out mice.

**Figure supplement 2.** Loss of TCAs in *Gbx2* knock-out mice does not affect neocortical layering at birth.

**Figure supplement 3.** Loss of TCAs in *Gbx2* knock-out mice does not affect neocortical arealization.

**Figure supplement 4.** Schematic of upper, middle and lower frames used for quantification of laminar distribution of GABAergic interneurons in newborn mouse neocortex.

**Figure supplement 5.** Normal proportion of GABAergic interneurons in superficial versus deep routes of tangential migration in *Gbx2* knock-out embryos.

**Figure supplement 6.** Lack of TCAs in *Gbx2* mutant mice affects the laminar distribution of different classes of MGE-derived interneurons to a similar extent.

other hand, more than twice as many GABAergic interneurons were found in the lower cortical sector of *Gbx2* knock-out mice compared to wild type controls (*Figure 1A and D*). Despite the differences between upper and lower layers, the combined counts of GABAergic interneurons across all cortical layers were not significantly different between wild type and *Gbx2* knock-out mice (*Figure 1E*). This indicated that, despite their abnormal layer distribution, MGE-derived GABAergic interneurons can reach the neocortex in normal numbers in the absence of TCAs. In agreement with this, there was no difference in cell proliferation in the E12.5 MGE of *Gbx2* knock-out embryos as assessed by BrdU labeling (data not shown). Neither was the total thickness of the cortical plate (from layer I to VI) different between the two genotypes at birth (data not shown). As GABAergic interneurons enter the cortical plate through both superficial and deep routes, we investigated whether the abnormal accumulation of interneurons in deeper cortical layers of *Gbx2* knock-out mice was due to a higher proportion of interneurons taking the deep route of tangential migration. However, the number of Lhx6-GFP$^+$ cells in superficial (MZ) versus deep (SVZ and IZ) routes was not different between wild type and *Gbx2* knock-out embryos (*Figure 1—figure supplement 5*), suggesting a defect in the radial dispersion of deep layer interneurons in the mutant.

Different lines of evidence have indicated a temporal bias in the generation of different subtypes of GABAergic interneurons (*Inan et al., 2012*; *Miyoshi et al., 2007*). It was therefore of interest to investigate whether early- and late-born GABAergic interneurons were equally affected by the loss of TCAs in *Gbx2* mutant mice. This was done by injecting pregnant females with BrdU at E12.5 and E14.5, respectively, and subsequently assessing the relative distribution of BrdU•Lhx6-GFP double positive cells in upper and lower cortical layers at E18.5. In both cases, *Gbx2* mutant mice showed increased numbers of MGE-derived interneurons in the lower layers of the cortex but fewer in the upper layers (*Figure 1—figure supplement 6A*), suggesting that both early- and late-born interneurons depend on TCAs to attain their normal laminar distribution. We also found that the relative proportion of Lhx6-GFP$^+$ cortical interneurons expressing the transcription factor *Satb1*, which is required for the survival of subsets of somatostatin- and parvalbumin-expressing cortical interneurons (*Close et al., 2012*; *Denaxa et al., 2012*), was not different in upper and lower cortical layers of *Gbx2* knock-out mice compared to wild type controls (*Figure 1—figure supplement 6B and C*), indicating that Satb1$^+$ cells were affected in a similar way as the total Lhx6-GFP$^+$ population in the mutants. Together, these data suggested that the lack of TCAs in the *Gbx2* mutant likely affected the laminar distribution of different classes of MGE-derived interneurons to a similar extent.

In order to further validate the effects of thalamic *Gbx2* on the distribution of cortical interneurons, we analyzed Lhx6-GFP$^+$ interneurons in the neocortex of *Olig3-Cre*$^{ERT2}$;*Gbx2*$^{fx/fx}$ conditional mutant mice. The *Olig3-Cre*$^{ERT2}$ allele directs expression of tamoxifen-regulatable CRE$^{ERT2}$ recombinase in the dorsal regions of the neural tube, including the developing thalamus, but not in the ventral telencephalon, where GABAergic interneurons are generated, or in the cortex (*Figure 2—figure supplement 1*) (*Storm et al., 2009*). *Olig3-Cre*$^{ERT2}$;*Gbx2*$^{fx/fx}$ mice injected with tamoxifen at E10.5 showed a significant reduction in *Gbx2* expression in the E12.5 thalamic anlage, as assessed by in situ hybridization (*Figure 2—figure supplement 2*). Using retrograde tracing, we could also verify a marked reduction of thalamo-cortical innervation in these mice compared to *Gbx2*$^{fx/fx}$ controls (*Figure 2—figure supplement 3*). In agreement with our observations in global *Gbx2* knock-out mice, MGE-derived interneurons were partially depleted from the upper cortical layers of *Olig3-Cre*$^{ERT2}$; *Gbx2*$^{fx/fx}$ conditional mutants, but were present in excess in lower layers (*Figure 2A–C*). It has previously been reported that *Gbx2* is expressed in a subset of MGE-derived *Lhx8*-positive cells that give rise to cholinergic interneurons in the striatum, but which do not contribute interneurons to the neocortex (*Chen et al., 2010*). Using *Gbx2*-Cre$^{ERT2}$;*dTom* reporter mice, we could also verify that Gbx2-expressing cells born during embryonic stages do not contribute neurons to the postnatal neocortex (*Figure 2—figure supplement 4*). In order to rule out any contribution of *Gbx2*-expressing MGE cells to the cortical phenotypes observed in *Gbx2* knock-out mice, we examined the distribution of Lhx6-expressing interneurons in the neocortex of *Nkx2.1*$^{Cre}$;*Gbx2*$^{fx/fx}$ conditional mutant mice, in which the *Gbx2* gene is deleted only in MGE-derived neuronal precursors. We found no abnormalities in either the complement or laminar distribution of MGE-derived interneurons in the neocortex of these mice (*Figure 2D,E*). We conclude from these studies that MGE-derived interneurons accumulate abnormally in deep layers of newborn mouse neocortex when this is deprived of thalamic innervation, suggesting a role for TCAs in the radial dispersion of cortical interneurons.

## Deficient tangential to radial orientation switch in GABAergic interneurons of neocortex lacking TCAs

The abnormal accumulation of interneurons in lower layers of the denervated neocortex suggested possible defects in their switching from a tangential to a radial orientation, a process that is required for their radial dispersion. In order to evaluate this, we assessed the proportion of radially oriented MGE-derived interneurons in upper and lower layers of M1, S1 and V1 primary cortical areas in *Gbx2* knock-out and wild type mice. GABAergic interneurons were considered to be in a radial orientation if their main process was at or less than a 25° angle from the radial axis of the cortex (*Martini et al., 2009*) (*Figure 3A*). Above this value, interneurons were considered to be in a tangential orientation. We found that fewer Lhx6-GFP$^+$ interneurons were oriented radially in the neocortex of *Gbx2* knock-out mice compared to wild type controls (*Figure 3B,C*). This was most pronounced in the lower cortical layers, where the proportion of radially oriented interneurons was reduced by over 50% in the mutant (*Figure 3C*) These data suggested that, in the absence of TCAs, excess GABAergic interneurons in the lower cortical layers remain in a tangential orientation. Changes in branching morphology of the leading process have been linked to tangential to radial migration switch in migrating MGE-derived GABAergic interneurons (*Baudoin et al., 2012*). We analyzed the morphology of the leading process in cortical areas of newborn *Gbx2* knock-out and wild type mice. We found significantly elevated frequency of highly branched leading processes in GABAergic interneurons of the mutants compared to wild type controls in both upper and lower cortical layers (*Figure 3D,E*). This type of morphology has been previously correlated with an inability to switch migration from a tangential to a radial orientation (*Baudoin et al., 2012*).

## Elevated KCC2 expression restrains the radial dispersion of GABAergic interneurons in neocortex lacking TCAs

TCAs could facilitate the radial dispersion of MGE-derived interneurons by providing physical support for radial migration. Alternatively, TCAs could supply signals that facilitate the tangential to radial orientation switch in GABAergic interneurons, thereby promoting their radial dispersion. The K$^+$/Cl$^-$ cotransporter KCC2 contributes to the hyperpolarizing influx of chloride ions which underlies the inhibitory function of GABA in the mature CNS (*Rivera et al., 1999*). KCC2 becomes upregulated in cortical interneurons as they mature, stop migrating and adopt their final position in the

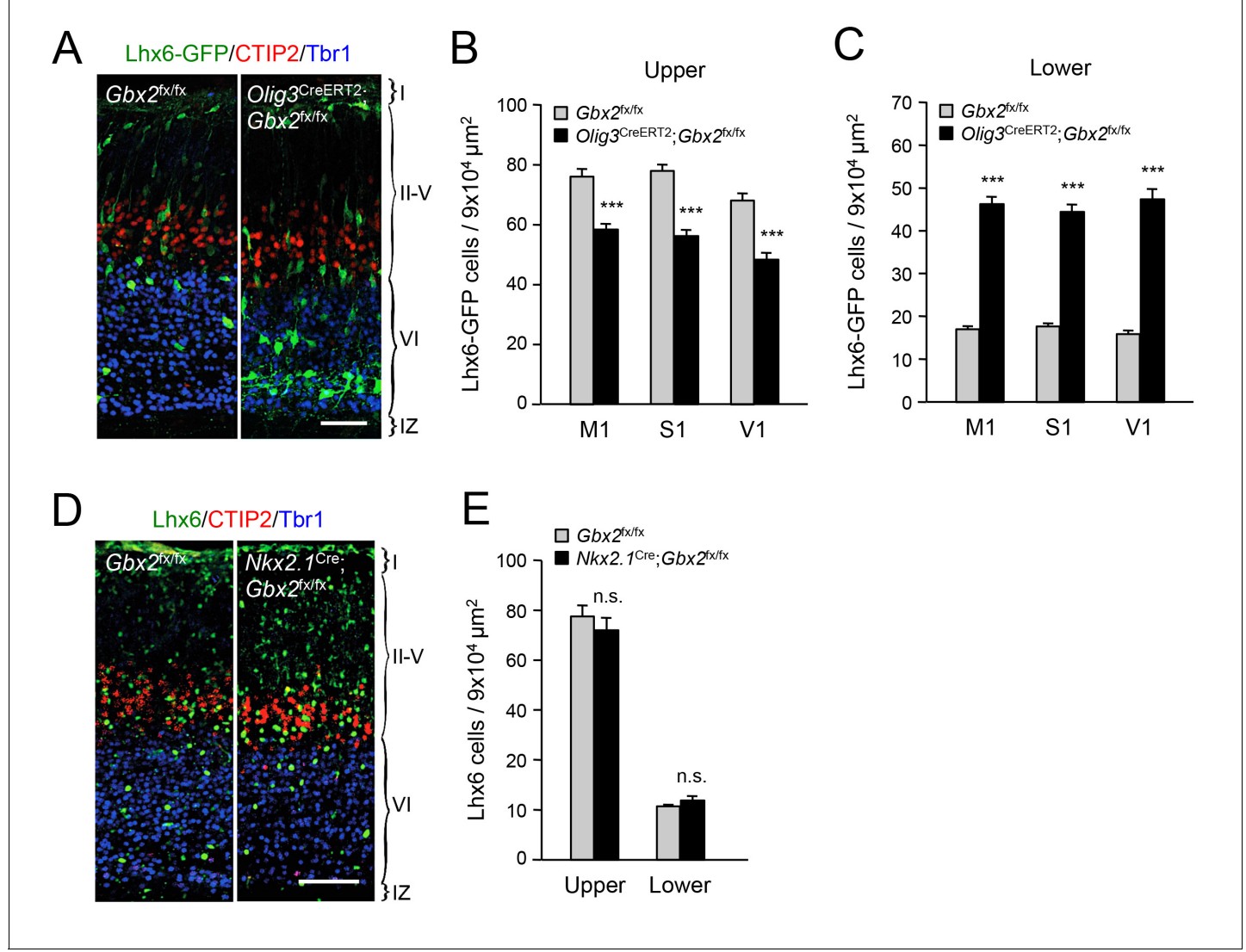

**Figure 2.** Thalamic *Gbx2* affects radial dispersion of cortical GABAergic interneurons non-cell-autonomously. (**A**) Lhx6-GFP[+] interneurons (green) combined with immunostaining for CTIP2 (red) and Tbr1 (blue) in prospective somatosensory cortex of newborn *Olig3-Cre*[ERT2]*;Gbx2*[fx/fx] conditional mutant and *Gbx2*[fx/fx] control mice after tamoxifen injection at E10.5. Scale bar, 50 μm. (**B–C**) Quantification of Lhx6-GFP[+] interneurons in upper (**B**) and lower (**B**) layers of prospective primary motor (M1), somatosensory (S1), and visual (V1) cortices of newborn *Olig3-Cre*[ERT2]*;Gbx2*[fx/fx] conditional mutant and *Gbx2*[fx/fx] control mice. Results are expressed as average ± SEM (***p<0.0005; N = 6 mice per group). (**D**) Lhx6[+] interneurons (green) combined with immunostaining for CTIP2 (red) and Tbr1 (blue) in prospective somatosensory cortex of newborn *Nkx2.1*[Cre]*;Gbx2*[fx/fx] conditional mutant and *Gbx2*[fx/fx] control mice. Scale bar, 100 μm. (**E**) Quantification of Lhx6[+] interneurons in upper and lower layers of prospective somatosensory cortex of newborn *Nkx2.1*[Cre]*;Gbx2*[fx/fx] conditional mutant and *Gbx2*[fx/fx] control mice. Results are expressed as average ± SEM (n.s., non-significant; N = 3 mice per group).

The following figure supplements are available for figure 2:

**Figure supplement 1.** Fate mapping of Olig3[+] E10.5 precursors at P0 and P21.

**Figure supplement 2.** Verification of removal of *Gbx2* expression in the thalamus of *Gbx2* mutant mice.

**Figure supplement 3.** Loss of TCAs in *Olig3-Cre*[ERT2]*;Gbx2*[fx/fx] conditional mutant mice.

**Figure supplement 4.** Fate mapping of Gbx2[+] precursor cells.

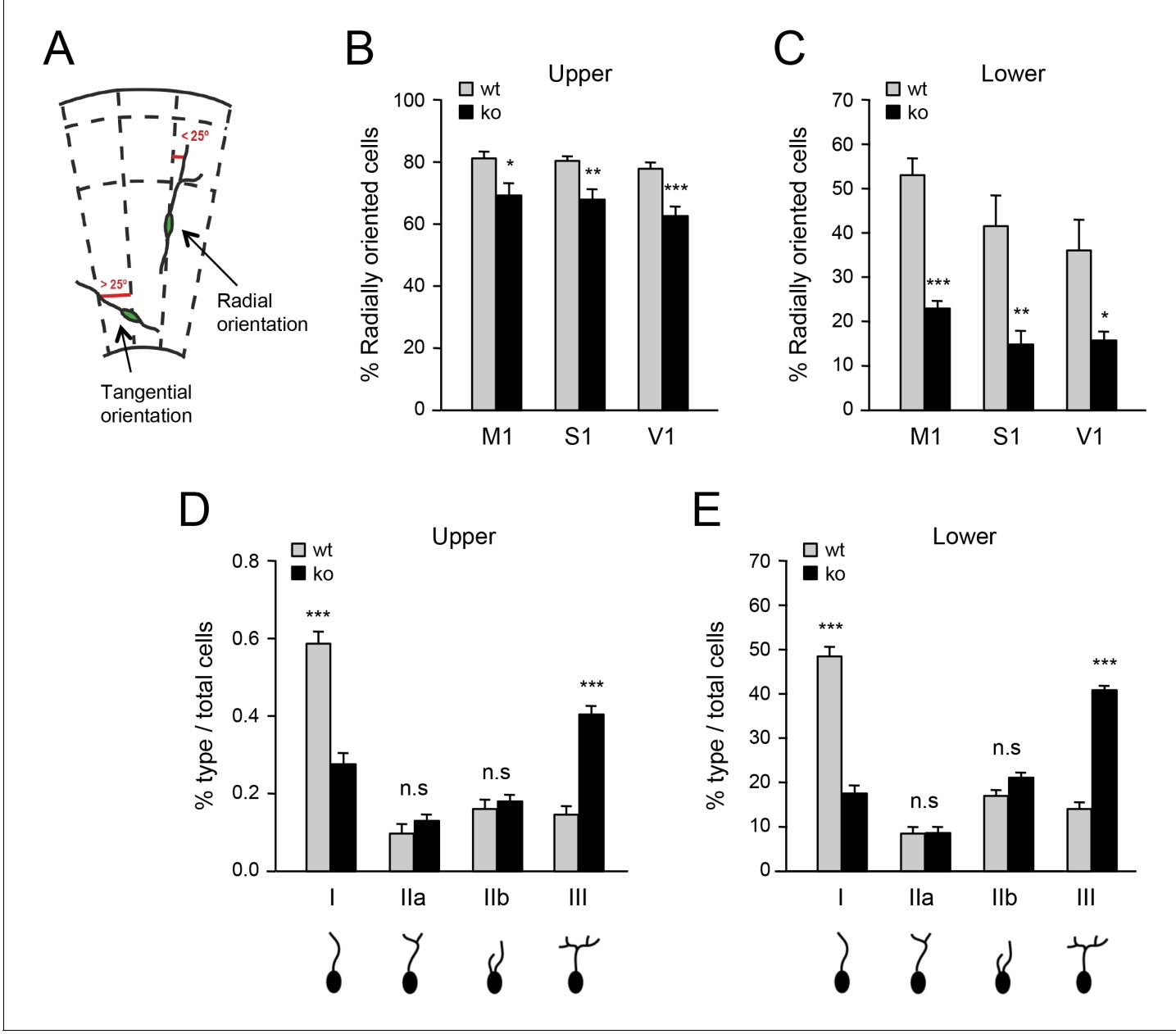

**Figure 3.** Deficient tangential to radial orientation switch in GABAergic interneurons of neocortex lacking TCAs. (A) Criteria for classification of the orientation of interneurons in cortical slices, modified from (*Martini et al., 2009*). (B–C) Quantification of the percentage of radially oriented Lhx6-GFP+ interneurons in upper (B) and lower (B) layers of prospective M1, S1, and V1 cortices of newborn wild type (wt) and *Gbx2* knock-out (ko) mice. Results are expressed as average ± SEM (*p<0.05; **p<0.005; ***p<0.0005; N = 3 mice per group). (D–E) Quantification of morphological types of Lhx6-GFP+ interneurons in upper (D) and lower (E) layers of prospective S1 cortex of newborn wild type (wt) and *Gbx2* knock-out (ko) mice. Results are expressed as average percentage of specific type of total interneuron number. ± SEM (***p<0.0001; n.s., not significant; N = 5 mice per group).

cortex. In vitro studies have indicated that upregulation of KCC2 is both necessary and sufficient to reduce interneuron motility (*Bortone and Polleux, 2009*), leading to the idea that KCC2 may function as a stop signal for migratory GABAergic interneurons once they reach their destination in the neocortex. We therefore investigated possible alterations in KCC2 expression among Lhx6-GFP+ interneurons in upper and lower cortical layers of *Gbx2* mutant and wild type neocortex. At birth, most KCC2-expressing cells were also positive for Lhx6-GFP (*Figure 4A*). Interestingly, the proportion of GABAergic interneurons expressing KCC2 was significantly elevated in the lower cortical

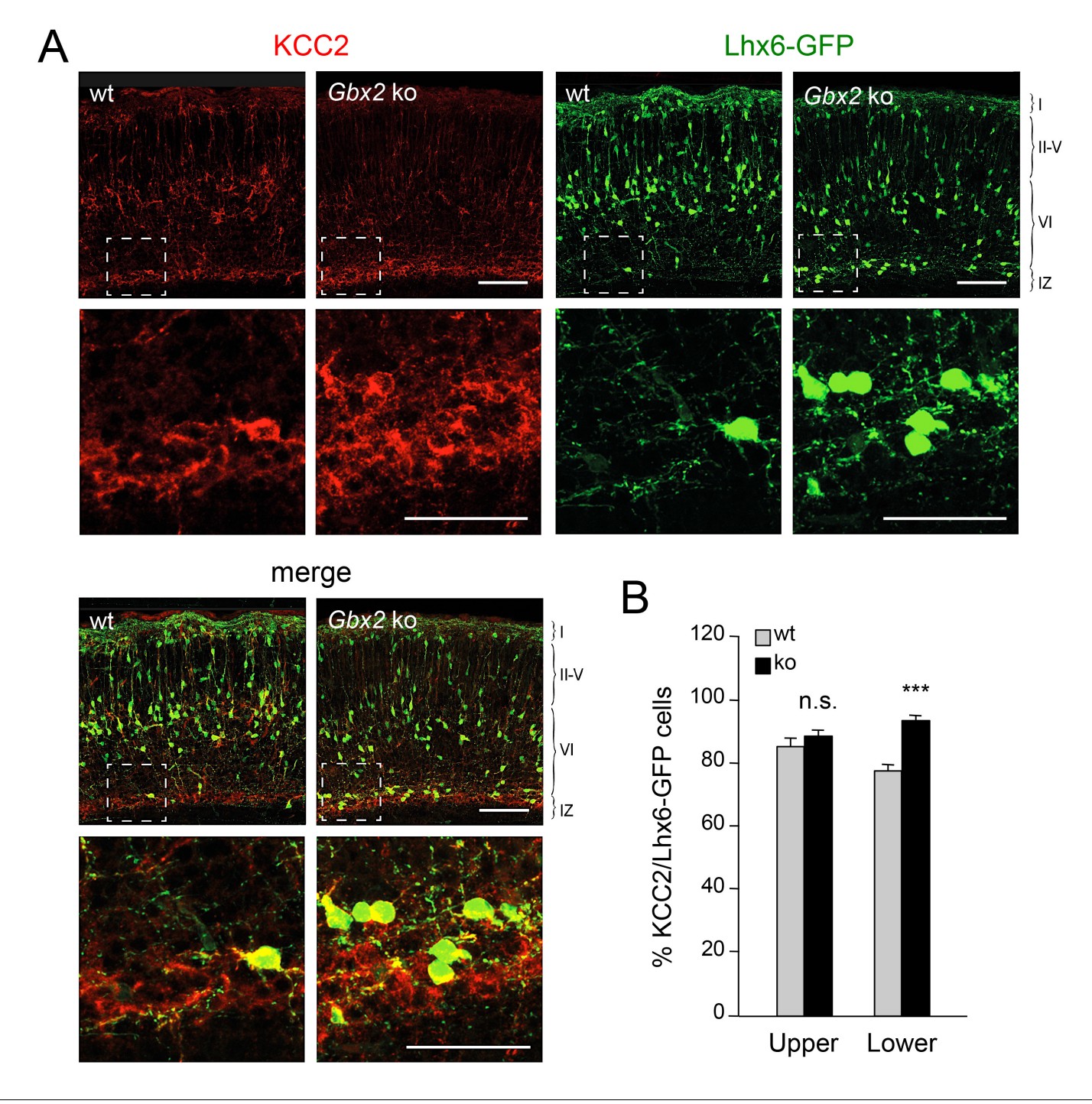

**Figure 4.** Elevated KCC2 expression in GABAergic interneurons of lower cortical layers lacking TCAs. (**A**) Expression of KCC2 (red) in Lhx6-GFP[+] interneurons (green) detected by immunohistochemistry in prospective somatosensory neocortex of newborn wild type (wt) and *Gbx2* knock-out (*Gbx2* ko) mice. Lower rows show higher magnification of area denoted in upper panels. Scale bar, 100 μm (upper rows), 50 μm (lower rows). (**B**) Quantification of the percentage of Lhx6-GFP[+] interneurons expressing KCC2 in upper and lower layers of wild type (wt) and *Gbx2* knock-out (ko) newborn mice. This shows that, not only do more Lhx6-GFP[+] interneurons accumulate in lower layers, but a greater proportion of these express KCC2. Results are expressed as average ± SEM (***p<0.0005; n.s., non-significant; N = 3 mice).

layers of *Gbx2* knock-out mice (*Figure 4B*). This suggested that abnormal upregulation of KCC2 expression may restrict the radial dispersion of interneurons in the *Gbx2* mutant. In order to test this notion, we examined the cerebral cortices of mutant mice lacking KCC2 and double mutants lacking both KCC2 and Gbx2. In the *Kcc2* mutant, we observed significantly more Lhx6-GFP$^+$ cells in the cortical plate at E16.5 compared to wild type embryos (*Figure 5A,B*), suggesting premature tangential to radial orientation switch and dispersion of GABAergic interneurons in the absence of KCC2. By E18.5, the number of GABAergic interneurons in upper cortical layers was comparable in wild type and mutant embryos (*Figure 5A,B*) and this was also maintained at birth (*Figure 5C,D*). As before, *Gbx2* knock-out mice presented excess GABAergic interneurons in lower cortical layers, but fewer in upper layers, compared to wild type controls (*Figure 5C,D*). Interestingly, this defect was rescued upon deletion of *Kcc2* in the double mutants (*Figure 5C,D*), supporting the notion that KCC2 restrains radial dispersion of MGE-derived GABAergic interneurons. We verified these results by specifically targeting KCC2 expression in MGE-derived GABAergic interneurons by *in utero* electroporation directed to the ventral telencephalon. For these experiments, we used a short-hairpin RNA targeting the *KCC2* mRNA (shKCC2) along with a construct encoding GFP under the control of a *Dlx5/6* promoter (Dlx5/6-GFP), to specifically mark prospective cortical GABAergic interneurons (*De Marco García et al., 2011*). Electroporation was performed at E13.5 in *Gbx2* knock-out and wild type embryos and the distribution of electroporated Dlx5/6-GFP$^+$ interneurons was analyzed at birth along with immunohistochemistry for KCC2 (*Figure 5E*). As expected, *Gbx2* knock-out embryos that received a control shRNA construct showed a higher accumulation of GABAergic interneurons in lower layers and lower numbers in upper layers compared to wild type controls (*Figure 5F*). In contrast, electroporation of shKCC2 normalized the distribution of GABAergic interneurons in the *Gbx2* mutants, eliminating their differences compared to wild type controls (*Figure 5G*); a result that was in agreement with the data obtained in double knock-out embryos. Together, these results suggested the possibility that TCA-derived signals may be responsible for limiting KCC2 levels in migratory GABAergic interneurons thereby facilitating their tangential to radial orientation switch and cortical invasion.

## TCA-derived glutamate limits KCC2 expression in GABAergic interneurons to facilitate their radial dispersion and cortical invasion

KCC2 levels can be downregulated by NMDA receptor (NMDAR) activity (*Lee et al., 2011*). Activation of NMDARs has been shown to induce proteolytic cleavage of KCC2 by the calcium-dependent protease calpain (*Puskarjov et al., 2012*; *Zhou et al., 2012a*). We speculated that TCA-derived glutamate, acting through the NMDAR/calpain pathway, could provide a mechanism to limit KCC2 levels in migratory GABAergic interneurons thereby allowing their radial dispersion in the cortical plate. In order to test the role of TCA-derived glutamate in the laminar dispersion of GABAergic interneurons, we used mice lacking the vesicular glutamate transporter VGLUT2 (also known as Slc17a6) in the thalamus, obtained by breeding *Vglut2*$^{\Delta/fx}$ mice (*Hnasko et al., 2010*; *Moechars et al., 2006*) to *Olig3-Cre*$^{ERT2}$ animals. It has been reported that both *Vglut1* (also known as Slc17a6) and *Vglut2* genes need to be inactivated to completely abolish glutamate release from TCAs in the adult mouse brain (*Li et al., 2013*). In newborn mice, however, we found that neither thalamic nuclei nor TCAs (as labeled by 5HTT immunostaining) nor neocortex express detectable levels of VGLUT1 (*Figure 6A*), a result that is in agreement with previous observations (*Kaneko and Fujiyama, 2002*; *Nakamura et al., 2005*). In contrast, we detected abundant expression of VGLUT2 in TCAs of the newborn mouse neocortex, which was completely abrogated in cortices of newborn *Gbx2* knock-out mice, confirming its TCA origin (*Figure 6B*). Importantly, VGLUT1 expression remained undetectable in TCAs from *Olig3-Cre*$^{ERT2}$;*Vglut2*$^{\Delta/fx}$ conditional mutant mice lacking VGLUT2 (*Figure 6—figure supplement 1*). At birth, wild type (*Vglut2*$^{+/+}$) and *Vglut2*$^{\Delta/fx}$ mice showed indistinguishable distribution of GABAergic interneurons in upper and lower layers of the neocortex (*Figure 6C*). In contrast, *Olig3-Cre*$^{ERT2}$;*Vglut2*$^{\Delta/fx}$ conditional mutant mice showed a phenotype similar to that of denervated cortex, namely elevated numbers of GABAergic interneurons in lower cortical layers but reduced in upper layers (*Figure 6C*). In addition, a greater proportion of GABAergic interneurons expressed KCC2 in conditional mutant mice lacking VGLUT2 in TCAs compared to both wild type and *Vglut2*$^{\Delta/fx}$ controls (*Figure 6D*). Together, these data support the idea that glutamate derived from TCAs restricts the expression of KCC2 in GABAergic interneurons to facilitate their radial dispersion and cortical invasion.

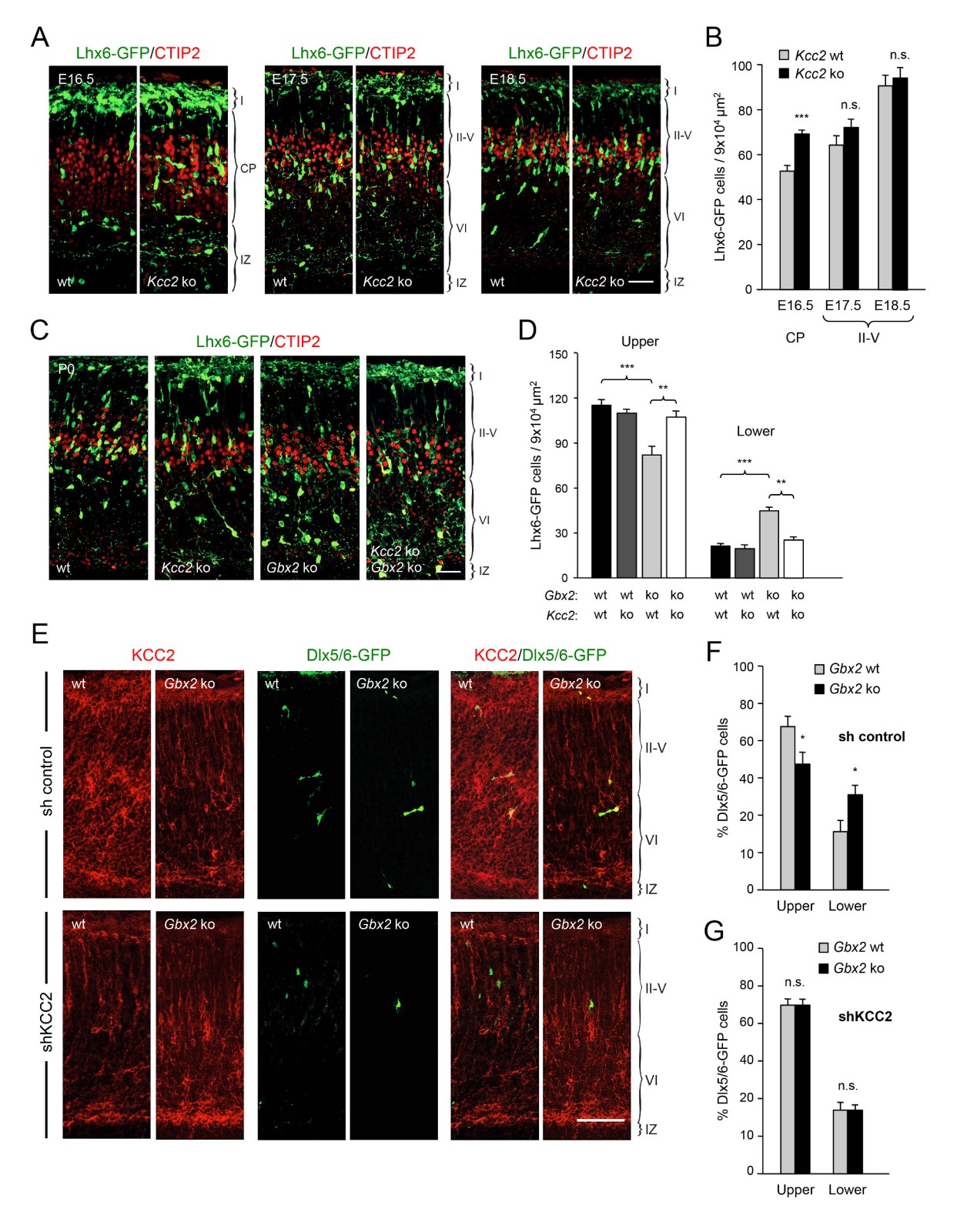

**Figure 5.** Deletion of *Kcc2* allows precocious radial dispersion of GABAergic interneurons in embryonic cortex and rescues abnormal laminar distribution of cortical interneurons in newborn *Gbx2* knock-out mice. (**A**) Lhx6-GFP[+] interneurons (green) combined with immunostaining for CTIP2 (red) in embryonic cortex of wild type (wt) and *Kcc2* knock-out (*Kcc2* ko) mice at embryonic stages E16.5, E17.5 and E18.5. Scale bar, 50 μm. (**B**) Quantification of Lhx6-GFP[+] interneurons in embryonic cortical plate (CP) or upper layers (II-V) of wild type (wt) and Kcc2 knock-out (*Kcc2* ko) mice at

*Figure 5 continued*

the indicated embryonic stages. Results are expressed as average ± SEM (**p<0.005; ***p<0.0005; N = 5 mice per group). (C) Lhx6-GFP+ interneurons (green) combined with immunostaining for CTIP2 (red) in prospective S1 cortex of newborn wild type (wt), *Kcc2* knock-out (*Kcc2* ko), *Gbx2* knock-out (*Gbx2* ko) and Kcc2/Gbx2 double knock-out (*Kcc2* ko *Gbx2* ko) mice. Scale bar, 50 μm. D) Quantification of Lhx6-GFP+ interneurons in upper and lower layers of newborn wild type mice and *Kcc2* and *Gbx2* single and double knock-out mice, as indicated. Results are expressed as average ± SEM (**p<0.005; ***p<0.0005; N = 4 mice per group). (E) Representative images of immunohistochemistry for KCC2 (red) of neonatal Gbx2 wildtype (wt) or mutant (*Gbx2* ko) cortex after in utero electroporation with control shRNA (first row) or KCC2 shRNA (second row) in combination with Dlx5/6GFP (green). Scale bar, 100 μm. (F–G) Quantification of GABAergic interneurons in upper and lower layers after *in utero* electroporation of either control shRNA (F) or KCC2 shRNA (G) in *Gbx2* wild type or knock-out embryos. Results are presented as percentage of transfected EGFP positive neurons in upper or lower cortical layers relative to all transfected cells. Results are expressed as average ± SEM (*p<0.05; N = 5; n.s., not significant difference).

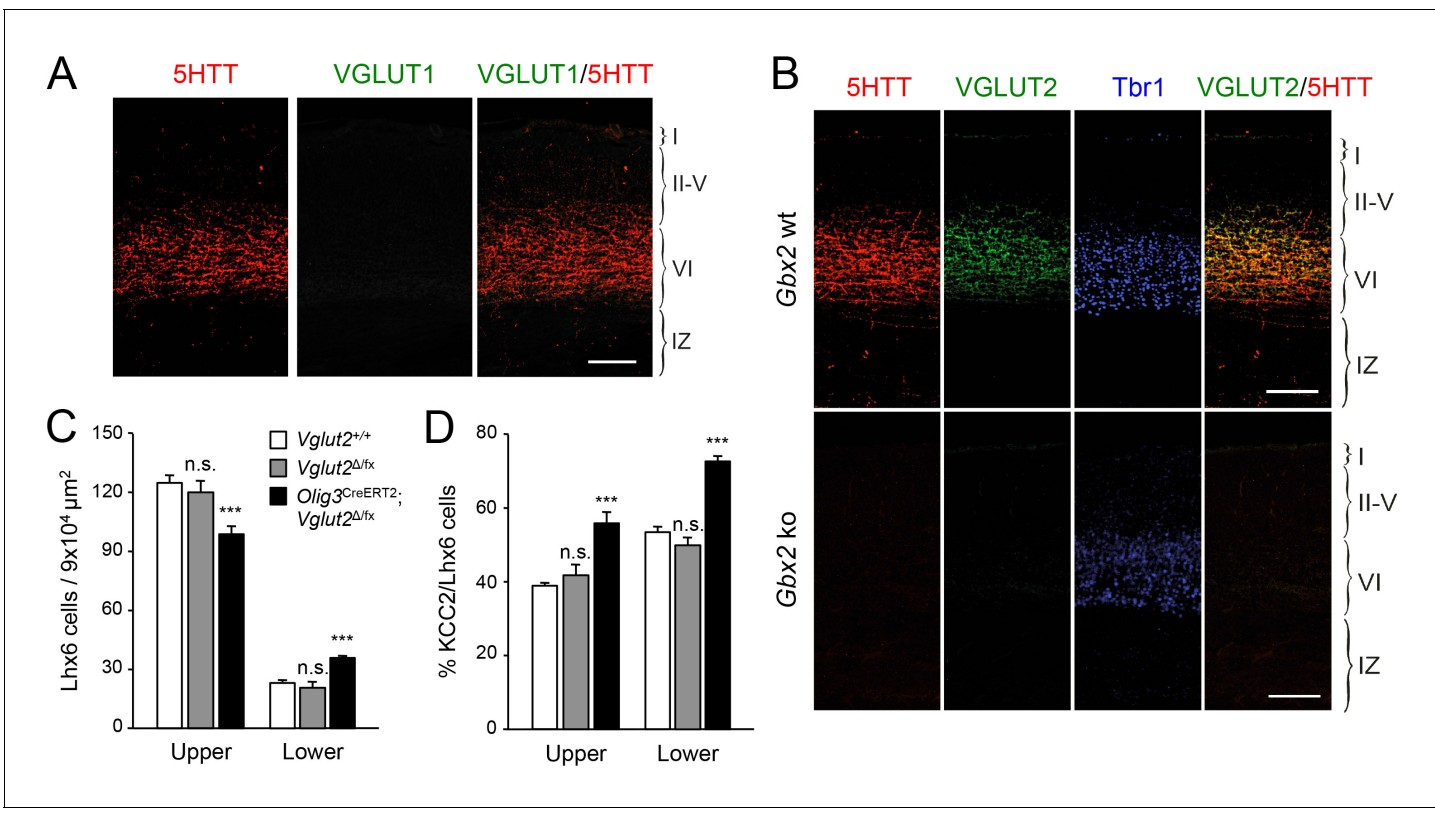

**Figure 6.** VGLUT2 ablation in TCAs induces abnormal laminar distribution of GABAergic interneurons in neonatal cortex. (A) Immunohistochemistry for VGLUT1 (green) and 5HTT (red) in prospective S1 cortex of wild type newborn mouse brain. Note the absence of VGLUT1 expression in TCAs (specifically labeled by 5HTT [*Mizuno et al., 2014*]) in the newborn neocortex. Scale bar, 100 μm. B) Immunohistochemistry for VGLUT2 (green), 5HTT (red) and Tbr1 (blue) in prospective S1 cortex of wild type and *Gbx2* knock-out newborn mouse brain. Note expression of VGLUT2 in TCAs (labeled by 5HTT) in the newborn wild type cortex and absence in the *Gbx2* knock-out mutant. Scale bar, 100 μm. (C) Quantification of Lhx6+ interneurons (identified by immunostaining) in upper and lower layers in newborn *Vglut2*+/+ (open bars), *Vglut2*Δ/fx (gray bars) and *Olig3-Cre*ERT2;*Vglut2*Δ/fx conditional mutant mice (black bars) after tamoxifen injection at E10.5. Results are expressed as average ± SEM (***p<0.0005; n.s., not significant; N = 5 mice per group). (D) Quantification of the percentage of KCC2-positive cells among Lhx6+ interneurons (identified by immunostaining) in upper and lower layers of newborn *Vglut2*+/+ (open bars), *Vglut2*Δ/fx (gray bars) and *Olig3-Cre*ERT2;*Vglut2*Δ/fx conditional mutant mice (black bars). Results are expressed as average ± SEM (***p<0.0005; n.s., non-significant; N = 3 mice).

The following figure supplement is available for figure 6:

**Figure supplement 1.** Absence of VGLUT1 expression in TCAs of *Olig3-Cre*ERT2;*Vglut2*Δ/fx conditional mutant mice.

## NMDA receptor subunit NR2B is required for normal radial dispersion and cortical invasion of GABAergic interneurons

The proposed role for NMDAR activity in the regulation of KCC2 expression (*Lee et al., 2011*; *Puskarjov et al., 2012*; *Zhou et al., 2012a*) prompted us to investigate the involvement of NMDAR subunits in the radial dispersion of GABAergic interneurons. Electrophysiological and expression studies have shown that, at birth, NMDARs in MGE-derived interneurons primarily contain the NR2B subunit (also known as Grin2b), with NR2A expression appearing at later stages (*Manent et al., 2006*; *Matta et al., 2013*; *Monyer et al., 1994*). Selective downregulation of NR2B expression in GABAergic interneurons was achieved by *in utero* electroporation directed to the ventral telencephalon. These experiments used two different short-hairpin RNAs targeting the *NR2B* mRNA (shNR2Bm and shNR2Bi), used together (m+i) or separately, along with the Dlx5/6-GFP to mark electroporated MGE-derived GABAergic interneurons, and a third construct to express red fluorescent protein (RFP) in all electroplated cells. The efficiency of shNR2Bm+i constructs was tested by assessing NR2B expression in cultures of MGE cells derived from embryos that were subjected to *in utero* electroporation (*Figure 7—figure supplement 1*). Expression of NR2B in Dlx5/6-GFP$^+$ MGE cells was down regulated by shNR2Bm+i but not by a control short hairpin RNA (*Figure 7—figure supplement 1*), confirming the efficacy of the shNR2B constructs. At birth, several RFP$^+$ cells could be seen in the neocortex of *in utero* electroporated mice, the majority of which also expressed Dlx5/6-GFP, indicating accurate targeting of the MGE during *in utero* electroporation (*Figure 7A*). Importantly, compared to the sh-control, MGE cells that received either of the two shNR2B constructs, or both combined, were fewer in upper cortical layers but more abundant in lower layers, similar to the phenotype observed in *Gbx2* and *Vglut2* mutant mice (*Figure 7B*). We investigated expression of KCC2 in GABAergic interneurons that had been depleted of NR2B by *in utero* electroporation and found a higher proportion of KCC2$^+$ cells compared to control shRNA (*Figure 7C and D*), in agreement with negative regulation of KCC2 expression by NMDAR activity. Together, these results indicated that NMDAR activity, and the NR2B subunit in particular, are required for radial dispersion and correct laminar distribution of cortical GABAergic interneurons by restricting the levels of KCC2 in these cells.

## Inhibition of calpain alters the laminar distribution of cortical GABAergic interneurons in wild type but not *Kcc2* mutant mice

Calpain is activated by NMDAR-mediated Ca$^{2+}$ influx and can proteolytically cleave KCC2 (*Puskarjov et al., 2012*; *Zhou et al., 2012a*). In order to address the involvement of calpain in the radial dispersion of cortical GABAergic interneurons, we administered the calpain inhibitor MDL28170 to pregnant females at E17.5 and E18.5 and assessed the laminar distribution of Lhx6-GFP$^+$ interneurons in newborn pups. We found that treatment with MDL28170 resulted in decreased numbers of GABAergic interneurons in upper cortical layers but increased numbers in lower layers (*Figure 8A,B*), mimicking the effects of TCA ablation in *Gbx2* mutant mice, loss of thalamic *Vglut2* and NR2B knock-down in MGE-derived interneurons. Treatment with MDL28170 also elevated the proportion of KCC2-expressing MGE-derived interneurons in neocortex compared to vehicle treatment (*Figure 8C*). Interestingly, MDL28170 had no effect on the laminar distribution of GABAergic interneurons in mice lacking KCC2 (*Figure 8D*), indicating that the effects of calpain on the radial dispersion of cortical GABAergic interneurons are dependent on KCC2.

## GABAergic interneuron deficits in postnatal neocortex lacking TCAs

In order to assess the longer term consequences of the absence of TCAs for cortical GABAergic interneurons, we examined the distribution of parvalbumin (PV) and somatostatin (SST) neurons, two major subpopulations of cortical GABAergic interneurons, in the postnatal neocortex of conditional *Olig3-Cre;Gbx2*$^{fx/fx}$ mutant mice. Unlike global *Gbx2* knock-out mice, *Olig3-Cre;Gbx2*$^{fx/fx}$ mutants survive postnatally, up to 5–6 weeks after birth (*Vue et al., 2013*). Similar to the *Olig3-Cre*$^{ERT2}$; *Gbx2*$^{fx/fx}$ strain, deletion of *Gbx2* in *Olig3-Cre;Gbx2*$^{fx/fx}$ mice is restricted to the thalamus and results in severe deficits in TCAs that persist into postnatal stages (*Vue et al., 2013*). However, unlike the former, *Olig3-Cre;Gbx2*$^{fx/fx}$ mice do not require tamoxifen for induction of Cre activity. At birth, *Olig3-Cre;Gbx2*$^{fx/fx}$ mice displayed lower numbers of MGE-derived Lhx6$^+$ GABAergic interneurons in the upper layers of the cortex but excess in the lower layers, in agreement with our observations

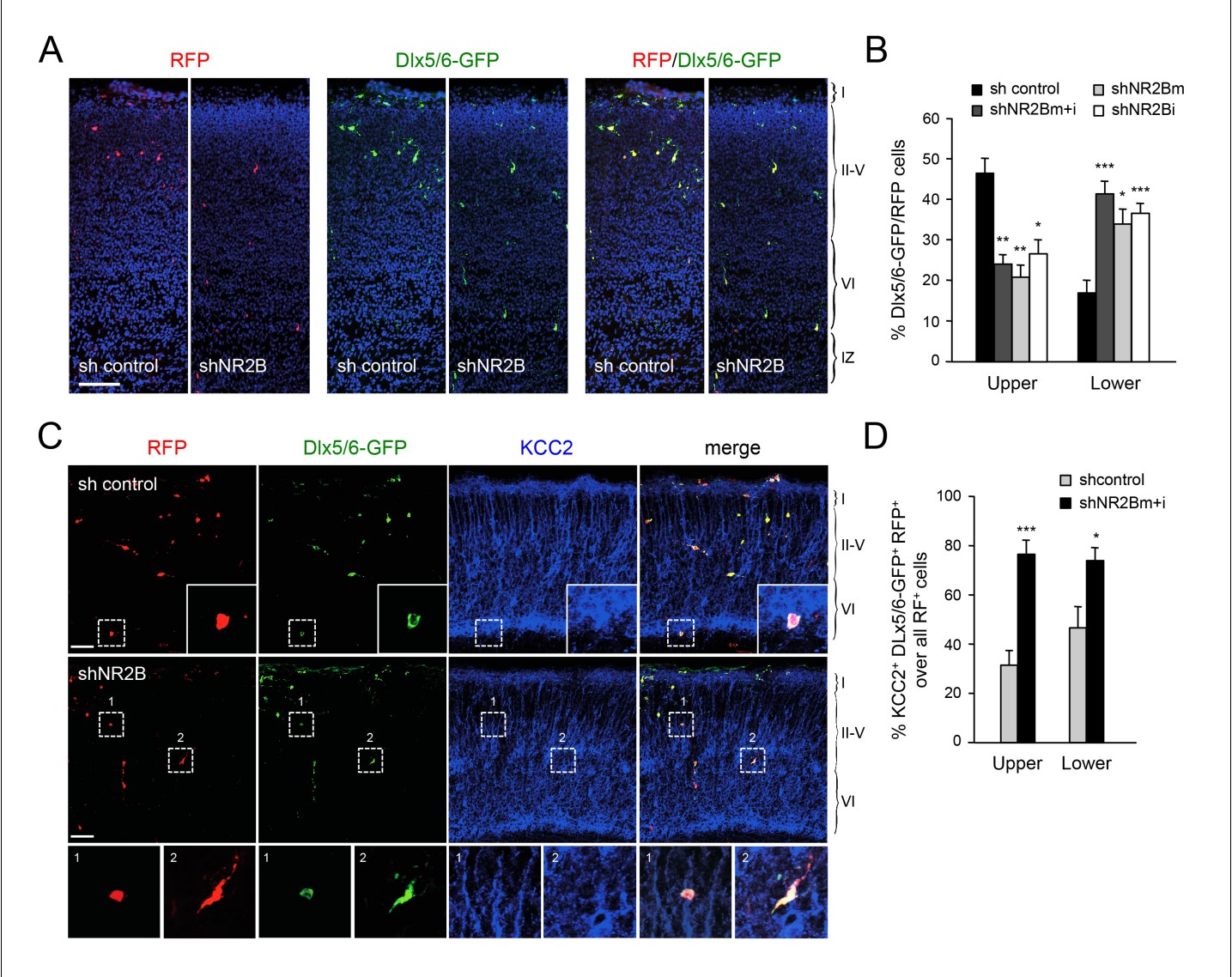

**Figure 7.** Abnormal laminar distribution of cortical GABAergic interneurons after interneuron-specific shRNA knock-down of NMDA receptor subunit NR2B. (A) GABAergic interneurons transfected by *in utero* electroporation with control or NR2B shRNAs combined with Dlx5/6-GFP (green) and RFP (red) from prospective S1 neocortex counterstained with DAPI (blue). RFP marks all transfected cells; Dlx5/6-GFP marks MGE-derived GABAergic interneurons. Note almost complete overlap between EGFP and RFP, indicating successful electroporation of MGE-derived interneurons. Scale bar, 100 µm. (B) Quantification of GABAergic interneurons in upper and lower layers after *in utero* electroporation of either control shRNA, NR2Bm shRNA, NR2Bi shRNA or combination of the two NR2B shRNAs (m+i). Results are presented as percentage of EGFP/RFP double-positive neurons in upper or lower cortical layers relative to EGFP/RFP double-positive neurons in all layers. Results are expressed as average ± SEM (*p<0.05; **p<0.005; ***p<0.0005; N = 13, sh control; N = 10, shNR2Bm+i; N = 9, shNR2Bm; N = 10, shNR2Bi). (C) GABAergic interneurons transfected by *in utero* electroporation with control or NR2B shRNAs combined with Dlx5/6-GFP (green) and RFP (red) from prospective S1 neocortex counterstained by immunohistochemistry for KCC2 (blue). RFP marks all transfected cells; Dlx5/6-GFP marks MGE-derived GABAergic interneurons. Scale bars, 50 µm. Insets show higher magnifications of areas inside dashed lines. (D) Percentage of KCC2$^+$ Dlx5/6-GFP$^+$ RFP$^+$ triple positive interneurons among all transfected (RFP$^+$) cells in upper and lower layers after *in utero* electroporation of either control shRNA (grey bars) or NR2B shRNA (black bars). Results are presented as percentage of triple-positive neurons in upper or lower cortical layers relative to all transfected neurons. Results are expressed as average ± SEM (**p<0.0001; *p<0.05; N = 4, sh control; N = 8, shNR2B).

The following figure supplement is available for figure 7:

**Figure supplement 1.** Knock-down of NMDA receptor subunit NR2B in GABAergic interneurons by *in utero* electroporation.

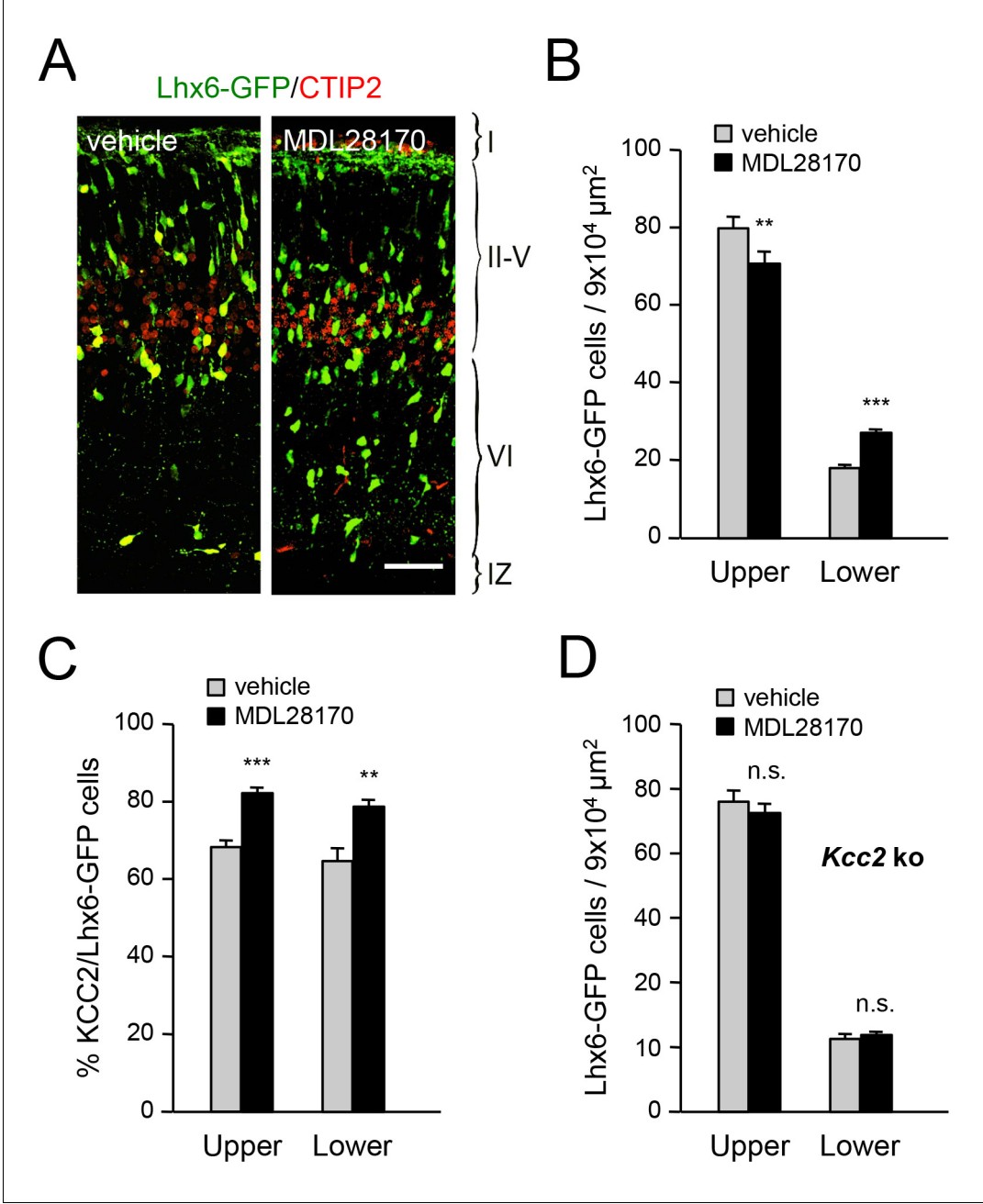

**Figure 8.** Inhibition of calpain alters the laminar distribution of cortical GABAergic interneurons in wild type but not in *Kcc2* mutant mice. (**A**) Lhx6-GFP[+] interneurons (green) combined with immunostaining for CTIP2 (red) in prospective S1 cortex of newborn mice treated at E17.5 and E18.5 with either vehicle or calpain inhibitor MDL28170. Scale bar, 50 μm. (**B**) Quantification of Lhx6-GFP[+] interneurons in upper and lower layers of prospective M1, S1 and V1 cortices from newborn wild type mice treated at E17.5 and E18.5 with either vehicle (gray bars) or calpain inhibitor MDL28170 (black bars). Results are expressed as average ± SEM (**p<0.005; ***p<0.0005; N = 5 mice per group). (**C**) Quantification of the percentage of KCC2-positive cells among Lhx6-GFP[+] interneurons in upper and lower layers of prospective M1, S1 and V1 cortices from newborn wild type mice treated at E17.5 and E18.5 with either vehicle or calpain inhibitor MDL28170. Results are expressed as average ± SEM (*p<0.05; **p<0.005; ***p<0.0005; N = 5 mice). (**D**) Quantification of Lhx6-GFP[+] interneurons in upper and lower layers of prospective M1, S1 and V1 cortices from newborn *Kcc2* knock-out mice treated at E17.5 and E18.5 with either vehicle (gray bars) or calpain inhibitor MDL28170 (black bars). Results are expressed as average ± SEM (n.s., not significant; N = 5 mice per group).

in the tamoxifen-inducible strain (*Figure 9—figure supplement 1*). Importantly, we detected a significant reduction in PV interneurons in layers II/III and IV of V1 cortex in three week old *Olig3-Cre*; *Gbx2*<sup>fx/fx</sup> mutant mice (*Figure 9A*). We also found significantly lower numbers of SST interneurons in cortical layers II/III of the mutants compared to control littermates (*Figure 9B*). These results indicate that the absence of TCAs results in permanent deficits in the normal complement of GABAergic interneurons in the neocortex.

## Discussion

Axon guidance and neuronal migration share common molecular and cellular mechanisms. They are exquisitely coordinated in space and time to ensure appropriate formation of neural circuits. The thalamocortical projection, one of the most prominent tracts in the forebrain, and MGE-derived interneurons, the farthest-reaching migratory cell population in the developing telencephalon, share a similar trajectory, directionality, site of cortical entry and timing of cortical invasion during development (*Bartolini et al., 2013*; *Lopez-Bendito and Molnár, 2003*; *Marín, 2013*). Whether MGE-derived interneurons derive guidance cues from TCAs for tangential migration and cortical invasion has been a matter of speculation but never directly demonstrated. Based mainly on in vitro studies, it was initially proposed that GABAergic interneurons may be guided by corticofugal axons in their migration towards the neocortex (*Denaxa et al., 2001*). However, later studies using mutant mice lacking those axons reached contradicting results, reporting either normal or partially reduced numbers of cortical GABAergic interneurons (*Ying et al., 2009*; *Zhou et al., 2010*). Mice lacking double-cortin (*Friocourt et al., 2007*) or the protocadherin Flamingo (*Ying et al., 2009*) have been reported to show defects in the cortical invasion of GABAergic interneurons accompanied by an overall reduction in cortex thickness. However, due to the wide expression of those proteins, both in GABAergic interneurons and many other cells, as well as the global nature of the mouse mutants employed, it remains unclear to which extent the effects observed were cell-autonomous or caused by other cortical abnormalities. For GABAergic interneurons derived from the CGE, the serotonin receptor 3A was recently found to play a role in cortical plate invasion (*Murthy et al., 2014*). Here we showed that MGE-derived interneurons can reach the neocortex in normal numbers in the absence of TCAs, indicating that they do not require this axonal tract for tangential migration from the ventral telencephalon nor for tangential dispersion through the subplate and IZ. However, we found that MGE-derived interneurons did require TCAs for proper radial dispersion and cortical invasion. Importantly, the *Gbx2* mutants used in this study showed no abnormalities in neocortical lamination, arealization or thickness. We further showed that signals derived from TCAs, such as glutamate, rather than the axons themselves, are required for the normal radial dispersion of MGE-derived interneurons. Mechanistically, we obtained genetic and pharmacological evidence supporting the cell-autonomous involvement of a NMDAR-calpain-KCC2 signaling cascade in this process.

### Thalamo-cortical axons regulate the radial dispersion of neocortical GABAergic interneurons

We have presented two lines of evidence demonstrating that the effects of *Gbx2* deletion on the laminar distribution of MGE-derived interneurons were caused non-cell-autonomously by disruption of the thalamocortical projection. First, conditional deletion of *Gbx2* in the dorsal telencephalon, including the thalamic nuclei but sparing the ganglionic eminences, altered the laminar distribution of MGE-derived interneurons in the same manner and to the same extent as the global ablation of *Gbx2*. Second, specific deletion of *Gbx2* in the MGE had no effect on the radial dispersion and laminar distribution of GABAergic interneurons in the neocortex. Our study of the orientation of the leading process of MGE-derived interneurons indicated that the switch from a tangential to a radial orientation is defective in the *Gbx2* mutant. In the mutant neocortex, many interneurons failed to align their leading processes in a radial orientation, including several neurons that had succeeded to enter the cortical plate, suggesting an incomplete tangential to radial switch. This switch requires branching and elongation of the leading process, as well as retraction of the non-preferred branch in its bifurcated tip (*Bellion et al., 2005*). Complete turning from a tangential to a radial position likely requires several cycles of leading process branching, branch selection, branch elongation and movement of the cell nucleus. In the *Gbx2* mutants, a higher proportion of interneurons remained in a tangential orientation with a highly branched leading process, a morphology that has previously been

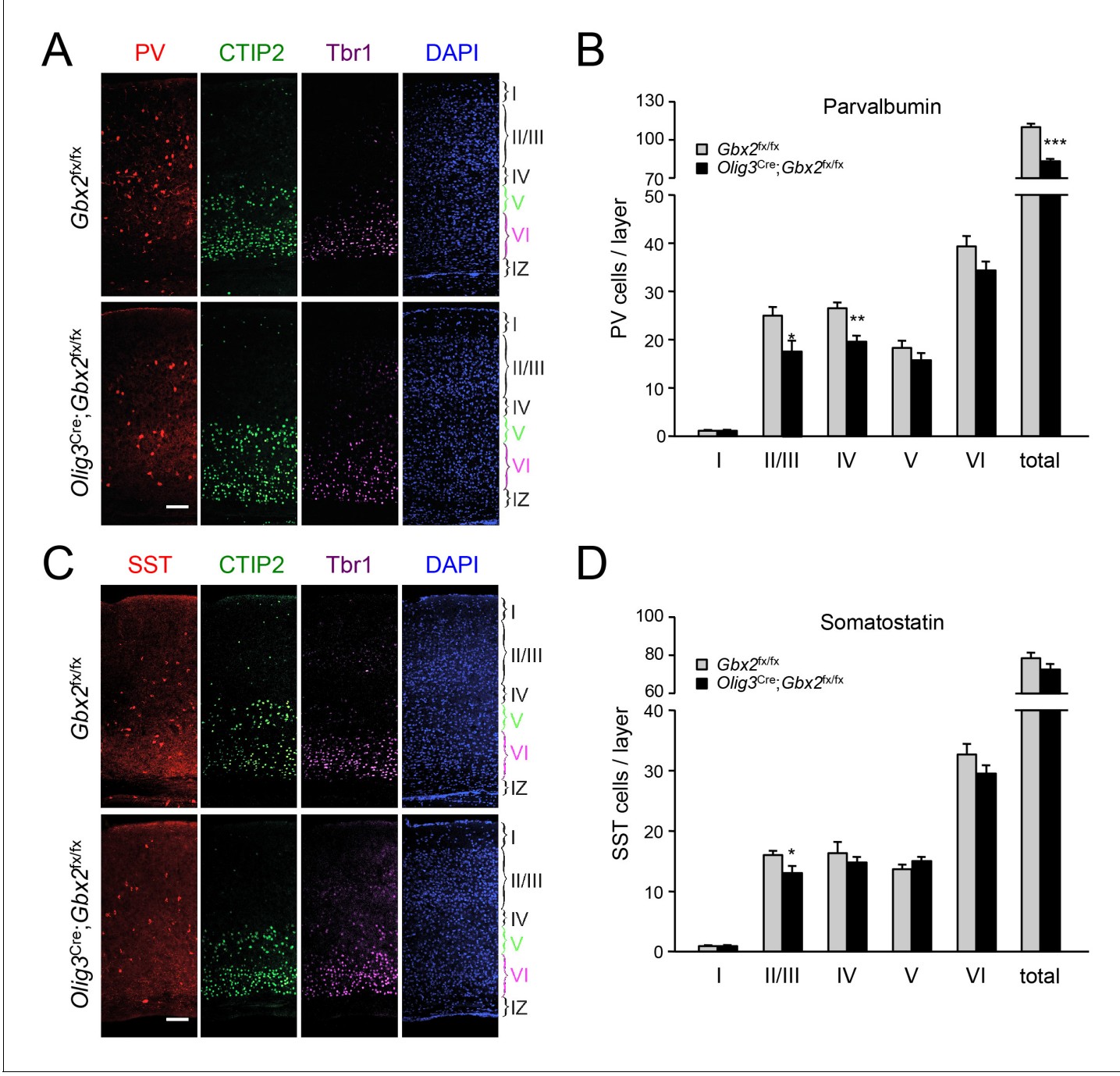

**Figure 9.** GABAergic interneuron deficits in postnatal neocortex lacking TCAs. (**A**) Immunostaining for parvalbumin (PV, red), CTIP2 (green), Tbr1 (purple) and DAPI counterstaining (blue) in primary visual cortex of 3 week old *Olig3-Cre;Gbx2*fx/fx conditional mutant and *Gbx2*fx/fx control mice. At this age, CTIP2 marks neurons in both layers V and VI. Cortical layers are indicated. Scale bar, 100 μm. (**B**) Quantification of PV⁺ in different layers of primary visual cortex in three week old *Olig3-Cre;Gbx2*fx/fx conditional mutant and *Gbx2*fx/fx control mice. Results are expressed as average ± SEM (*p<0.05; **p<0.005; ***p<0.001; N = 6 mice per group). (**C**) Immunostaining for somatostatin (SST, red), CTIP2 (green), Tbr1 (purple) and DAPI counterstaining (blue) in primary visual cortex of 3 week old *Olig3-Cre;Gbx2*fx/fx conditional mutant and *Gbx2*fx/fx control mice. Cortical layers are indicated. Scale bar, 100 μm. (**D**) Quantification of SST⁺ in different layers of primary visual cortex in three week old *Olig3-Cre;Gbx2*fx/fx conditional mutant and *Gbx2*fx/fx control mice. Results are expressed as average ± SEM (*p<0.05; N = 6 mice per group).

The following figure supplement is available for figure 9:

**Figure supplement 1.** Abnormal laminar distribution of cortical GABAergic interneurons in newborn *Olig3-Cre;Gbx2*fx/fx mice.

linked to abnormal tangential to radial orientation switch in MGE-derived interneurons (**Baudoin et al., 2012**).

## KCC2 negatively regulates the radial dispersion of neocortical GABAergic interneurons

The precise molecular mechanisms underlying the migration of cells with a bifurcated leading process are not well understood (**López-Bendito et al., 2008**). KCC2 expression increases upon maturation of cortical GABAergic interneurons and has been shown to function as a stop signal for the migration of these cells in in vitro studies (**Bortone and Polleux, 2009**). This effect was proposed to be mediated by KCC2's ability to reduce membrane potential upon activation of GABA$_A$ receptors, thereby decreasing the activity of voltage sensitive calcium channels (VSCCs) and overall Ca$^{2+}$ dynamics in the cell (**Bortone and Polleux, 2009**). KCC2 has also been shown to interact with the actin cytoskeleton to regulate spine formation, an activity that appears to be independent from its chloride transport function (**Li et al., 2007**; **Llano et al., 2015**). Further work will be required to determine whether KCC2 regulates tangential to radial switch and laminar dispersion of cortical GABAergic interneurons through its effects on GABA signaling, the cytoskeleton or some other mechanism. In either case, high KCC2 levels may slow down the branching cycles of the leading process of GABAergic interneurons, resulting in an insufficient number of cycles for proper tangential to radial turning. In the *Kcc2* knock-out, a higher frequency of branching cycles may explain the premature radial dispersion of GABAergic interneurons that we observed at E16.5. Deletion or knockdown of *Kcc2* rescued the radial dispersion phenotype of MGE-derived interneurons in the neocortex of *Gbx2* knock-out mice. As the neocortex in these double mutants is also expected to lack TCAs, it can be concluded that neither tangential to radial orientation switch nor radial dispersion of cortical GABAergic interneurons require the actual presence of TCAs in the cortex. Thus, GABAergic interneurons do not appear to require TCAs as guide or substrate for laminar invasion. Instead, our results indicate that radial dispersion of GABAergic interneurons requires signals derived from TCAs and that this requirement can be circumvented by depletion of KCC2.

## TCA-derived glutamate facilitates cortical invasion of GABAergic interneurons through regulation of KCC2 levels

It has been reported that KCC2 has a high turnover in neurons, being significantly or totally replaced within 10 min (**Lee et al., 2007**; **Rivera et al., 2004**). The ability of the NMDA receptor to regulate KCC2 levels (**Lee et al., 2011**; **Puskarjov et al., 2012**; **Zhou et al., 2012a**) indicated the possibility that glutamate derived from TCAs regulates KCC2 levels in cortical GABAergic interneurons at the time of their tangential to radial switch. Our analysis of *Vglut2* conditional mutant mice brings support to this idea. Vglut2 was found to be the main vesicular glutamate transporter in TCAs of newborn mice. Ablation of *Vglut2* in thalamic nuclei phenocopied *Gbx2* mutants in both the abnormal laminar distribution of and increased KCC2 expression in cortical GABAergic interneurons. We propose that the release of glutamate by TCAs in the environment of the subplate and IZ limits KCC2 expression in migratory GABAergic interneurons, facilitating completion of tangential to radial orientation switch and cortical invasion. By restricting KCC2 levels in migrating interneurons, TCA-derived glutamate may not only contribute to coordinate TCA and interneuron cortical invasion but also allow interneurons to reach their final laminar position in the neocortex before their full maturation.

## Role of the NMDAR-calpain-KCC2 cascade in the control of GABAergic interneuron migration into the neocortex

The discovery of the NMDAR-calpain-KCC2 cascade (**Lee et al., 2011**; **Puskarjov et al., 2012**; **Zhou et al., 2012a**) offered a plausible molecular mechanism for the control of interneuron KCC2 levels. Interneurons express NMDAR subunits (**Manent et al., 2006**; **Matta et al., 2013**; **Monyer et al., 1994**) and NMDAR activation increases their intracellular Ca$^{2+}$ concentration and motility (**Bortone and Polleux, 2009**; **Soria and Valdeolmillos, 2002**). In our studies, we found that downregulation of the NR2B subunit in MGE-derived interneurons hinders their radial dispersion, in agreement with a cell-autonomous role of NMDAR signaling in the regulation of cortical invasion of GABAergic interneurons. NMDAR activity and calcium influx have been proposed to restrict KCC2 levels through proteolytic cleavage by the calcium-activated protease calpain (**Puskarjov et al.,**

*2012*; *Zhou et al., 2012a*). The human genome has revealed over a dozen calpains, with calpain isoforms 1 and 2 being the most abundant in the nervous system (*Liu et al., 2008*). Calpain substrates include many cytoskeletal and signaling proteins and there is abundant evidence linking calpains to the regulation of cell migration, although to date mainly in non-neuronal cells (*Franco and Huttenlocher, 2005*). Calpain activity has been shown to contribute to the regulation of leading process branching in cultured cortical interneurons (*Lysko et al., 2014*), one of the key steps in tangential to radial orientation switch. Our studies show that pharmacological blockade of calpain increases KCC2 expression in cortical GABAergic interneurons and hampers the radial dispersion. Importantly, inhibition of calpain had no effect on the laminar distribution of GABAergic interneurons in *Kcc2* mutant mice, indicating that the ability of calpain to affect migratory interneurons is dependent on KCC2. These findings provide the first demonstration of a role for calpain in the radial dispersion of interneurons in vivo.

## GABAergic interneuron deficits in postnatal neocortex lacking TCAs

Deficits in PV and SST interneurons in the neocortex of 3 week old conditional *Olig3-Cre;Gbx2*fx/fx mice indicate that the absence of TCAs did not simply cause a delay in radial dispersion of GABAergic cells but resulted in permanent deficits in the normal complement of cortical interneurons. It is possible that such deficits are the direct consequence of the initial interneuron deficit in the upper cortical layers of newborn mutants. It should also be noted that, after invading the cortical plate by radial migration, GABAergic interneurons continue sorting into specific cortical layers during the first days after birth; by 3 weeks of age, their laminar positioning is complete (*Miyoshi and Fishell, 2011*). It is therefore possible that this sorting process is altered in the mutants due to the lack of TCA input. In the deeper cortical layers, the normal complement of PV and SST interneurons in three week old mutants suggests that the initial excess of interneurons in the lower layers was subsequently eliminated, or that the excess cells did not differentiate into mature PV and SST cells. We note that we have not detected increased cell death, as assessed by immunostaining for cleaved caspase-3, in neocortex of newborn or 3-week old *Olig3-Cre;Gbx2*fx/fx mice (data not shown), indicating that any cell loss should have occurred between P0 and P21. Further work will be needed to elucidate how the abnormal distribution of GABAergic interneurons in the newborn cortex lacking TCAs results in layer-specific deficits in the mature cortex.

## Conclusions

Important advances have been made in recent years on our understanding of the mechanisms controlling tangential migration of cortical GABAergic interneurons from the ganglionic eminences to the neocortex (*Marín, 2013*). In comparison, the subsequent stages in this process, including the switch from a tangential to a radial orientation and the radial dispersion of interneurons in the neocortex, are much less understood (*Bartolini et al., 2013*). This study provides new insights into the mechanisms controlling these processes, and presents a new rationale for the coordinated radial invasion of the neocortex by MGE-derived interneurons and TCAs.

# Materials and methods

## Animals

*Lhx6-GFP* mice carry a modified BAC with the EGFP reporter gene inserted immediately upstream of the coding sequence of the *Lhx6* gene (*Gong et al., 2003*) and were obtained from the Mutant Mouse Regional Resource Centers (MMRRC) (stock 000246-MU). *Gbx2* knock-out mice (*Wassarman et al., 1997*) and *Gbx2*fx mice (*Li et al., 2012*) were provided by James Y. H. Li (University of Connecticut Health Center, Farmington, USA). *Nkx2.1*Cre mice (*Xu et al., 2008*) were provided by Jens Hjerling-Leffler (Karolinska Institute, Stockholm, Sweden) and *Gbx2*-CreERT2 mice (*Chen et al., 2009*) by Juha Partanen (University of Helsinki, Helsinki, Finland). *Olig3-Cre*ERT2 mice (*Storm et al., 2009*) were provided by Carmen Birchmeier (Max-Delbrück-Centrum for Molecular Medicine, Berlin, Germany). *Olig3-Cre* mice were generated as previously described (*Vue et al., 2009*). *Kcc2* knock-out mice (*Hübner et al., 2001*) were provided by Kai Kaila (University of Helsinki, Helsinki, Finland). *vGlut2*fx (*Hnasko et al., 2010*) and *vGlut2* knock-out (referred as *Vglut2*Δ) mice (*Moechars et al., 2006*) were provided by Ole Kiehn (Karolinska Institutet, Stockholm, Sweden).

dTomato (dTom) reporter mice (*Madisen et al., 2010*) were purchased from Charles River. Neonatal pups were collected as newborns in all cases, except for *Gbx2⁻/⁻;Kcc2⁻/⁻* double mutant mice, which were collected by caesarean section at E19. Animal protocols were approved by Stockholms Norra Djurförsöksetiska nämnd and are in accordance with the ethical guidelines of the Karolinska Institute.

## Immunohistochemistry

Embryos and neonatal pups were decapitated and fixed in 4% paraformaldehyde (PFA, Sigma, St. Louis, Missouri, USA) for 24 hr at 4°C. Three week old mice were deeply anesthetized and intracardially perfused with PBS followed by 4% PFA and postfixed for 24 hr at 4°C. After cryoprotection in 30% sucrose overnight, 20 μm coronal sections were collected on a cryostat. For Lhx6 and Tbr1 immunohistochemistry, antigen retrieval was performed by boiling sections in citrate buffer for 5 min prior to immunostaining. Lhx6-GFP reporter expression was detected by immunohistochemistry with anti-GFP antibodies. The antibodies utilized were as follows: goat anti-EGFP (1:500, ab6673, Abcam, Cambridge, UK), goat anti-Lhx6 (1:50, H75, sc-98607, 1:50, Santa Cruz, Dallas, Texas, USA), rat anti-CTIP2 (1:500, ab18465, Abcam, Cambridge, UK), rabbit anti-5HTT (1:500, PC177L, Calbiochem, San Diego, USA), mouse anti-NMDAR2B (1:200, 610416, BD Biosciences, Franklin Lakes, New Jersey, USA), rabbit anti-KCC2 (1:500, provided by Claudio Rivera, University of Helsinki, Finland, and from Millipore, 07–432), mouse anti-KCC2 (N1/12, AB_10672851, UC Davis/NIH NeuroMab, Davis, California, USA), mouse anti-parvalbumin (1:500, PV235, Swant, Switzerland), rabbit anti-somatostatin (1:500, T-4103, Penninsula laboratories, San Carlos, California,USA), goat anti-SATB1 (1:500, sc5989X, Santa Cruz, Dallas, Texas, USA), chicken anti-Tbr1 (1:500, AB2261, Millipore, Billerica, Massachusetts, USA), guinea pig anti-vGlut1 (1:500, AB5905, Millipore, Billerica, Massachusetts, USA) and guinea pig anti-vGlut2 (1:500, AB2251, Millipore, Billerica, Massachusetts, USA). Following washing in PBS and 2 hr incubation with secondary antibody (1:500; AlexaFlour 488, 568 or 649, Molecular Probes, Eugene, Oregon, USA), sections were counterstained with 4′,6-Diamidino-2-phenylindol (DAPI, 1:10000, Molecular Probes, Eugene, Oregon, USA) for 10 min, washed in PBS and embedded in fluorescence mounting medium (DAKO, Glostrup, Denmark).

## BrdU labeling and DiI tracing

For BrdU labeling, time-mated females received one intraperitoneal (i.p.) injection with BrdU (100 mg/kg; Sigma, St.Louis, Missouri, USA,) at either 12.5 days post coitum (d.p.c.) or E14.5. Embryos were collected at the indicated times after BrdU administration in 4% PFA. After cryoprotection in 30% sucrose overnight, cryostat sections were incubated in 2M HCl for 1 hr at 37°C to denature the DNA. BrdU was detected by immunohistochemistry with rat anti-BrdU (1:500, 347580, AbD Serotec, Hercules, California, USA, [*Xu et al., 2010*]). For tracing of neonatal brains, the tissue was fixed in PFA and DiI crystals (Molecular Probes, Eugene, Oregon, USA) were placed into either the visual or the somatosensory cortices using an insect needle. For anterograde tracing of TCAs at E16.5, brains were dissected along the midline and DiI crystals were placed in the dorsal thalamus. After incubation at 37°C for three weeks, the tissue was cut on a freezing microtome (30 μm, Leica, Nussloch, Germany), mounted on slides and stained with DAPI as described above.

## Riboprobe synthesis and in situ hybridization

Riboprobe synthesis and in situ hybridization followed by immunohistochemistry were performed as described in (*Zechel et al., 2014*). Riboprobes were derived from DNA fragments obtained by PCR from neonatal cortical cDNA using the following primers: Cux1 (fw: CTCAGAAAGCACTCCAAAGACC; rev: CTTCCAGCTTGAATCTCCTCAA); CTGF (fw: AAATGCTGCGAGGAGTGG; rev: TGTGCGTTCTGGCACTGT); Lmo4 (fw: GCTCCCTCTCCTGGAAGC; rev: GGGGCCCTGCTAATTGTT); RORß (fw: ACCTGAACACCGAGACCG; rev: CCCTTCATTTGCAGACCG). Riboprobes for Cad8 was supplied by Christoph Redies, University Hospital, Jena, Germany (*Korematsu and Redies, 1997*), for Lhx6 by Vassilis Pachnis, Francis Crick Institute, London, UK, and for Gbx2 by Alex Joyner, Memorial Sloan Kettering Cancer Center, New York, USA (*Bouillet et al., 1995*). All riboprobes were verified by DNA sequencing. In situ hybridizations on tissue samples were repeated several times including a sense control for each individual riboprobe.

## In utero electroporation and cell culture

*In utero* electroporation was performed as described (*Ngô-Muller and Muneoka, 2010*; *Tabata and Nakajima, 2008*) on pregnant dams carrying E13.5 embryos by injecting shNR2Bi and shNR2Bm plasmids, separately or together, for knockdown of NR2B expression (*Zhou et al., 2012b*) (provided by Hongbing Wang, Michigan State University, East Lansing, Michigan, USA), each at 1 µg/µl. In a second set of experiments, knockdown of KCC2 was achieved with shKCC2 plasmid (*Bortone and Polleux, 2009*) (provided by Franck Polleux, University of North Carolina, Chapel Hill, North Carolina, USA), at 1 µg/µl. The shRNA sequences were as follows: NR2Bi: 5′-GCGCATCATCTCTGAGAA TAA-3′, NR2Bm: 5′-GGATGAGTCCTCCATGTTC-3′; shKCC2: 5-AGCGTGTGACAATGAGGAGAA-3′; and sh control: 5′-ACTACCGTTGTTATAGGTGT-3′. RFP expression plasmid (*Campbell et al., 2002*) (1 µg/µl; provided by Masanori Uchikawa, University of Osaka, Osaka, Japan) and Dlx5/6-EGFP plasmid (1 µg/µl; provided by Gord Fishell, New York School of Medicine, New York, USA) were applied together with shRNAs and combined with 0.05% Fast green (Sigma, St.Louis, Missouri, USA) as a tracer. We note that the Dlx5/6-EGFP construct predominantly marks MGE-derived interneurons when injected at this age (I.e. E13.5 and earlier). At later stages, i.e. E15.5 and later, Dlx5/6-EGFP will also mark CGE-derived interneurons, as described previously (*De Marco García et al., 2011*). For electroporation, tweezer electrodes were placed horizontally on both hemispheres of the embryo with the anode tilted towards the lower jaw of the embryo on the side of the injected hemisphere in order to preferentially target the MGE in the ventral telencephalon. The square wave electroporator NEPA21 (Nepagene, Chiba, Japan) was used to deliver two 30 ms pulses of 50 V for poring and five 50 ms pulses of 35 V for transferring (both with 450 ms intervals) to each embryo. Neonatal brains were postfixed in 4% PFA at 4°C overnight and processed for immunohistochemistry as described above. To verify the efficiency of electroporation, MGEs of E13.5 embryos were dissected directly after the *in utero* electroporation, dissociated and plated on laminin/polylysin-coated glass cover slips in Neurobasal medium (Gibco, Thermofisher, Waltham, Massachusetts, USA), supplemented with B27 and glutamine, and maintained in culture for four days. The cells were then fixed in 4% PFA and stained as described above.

## Drug administration

Time-mated females received one intraperitoneal (i.p.) injection with tamoxifen (100 mg/kg in corn oil; Sigma, St. Louis, Missouri, USA) at 10.5 d.p.c. to induce recombination of the *Gbx2*fx allele in *Olig3*-CreERT2;*Gbx2*fx/fx embryos or activation of dTom in *Gbx2*-CreERT2;*dTom* embryos. Morning vaginal plug was considered 0.5 d.p.c., or E0.5 (for embryo staging). For treatments with calpain inhibitor, time-mated females received two injections at 17.5 and 18.5 d.p.c. with either saline solution (0.9%, Braun, Melsungen, Germany) or calpain inhibitor MDL28170 (240 µg/kg in 100 mM DMSO; Tocris Bioscience, Bristol, UK).

## Image analysis

Immunofluorescence images were taken with a Carl Zeiss LSM710 confocal microscope (10 µm thick, 20 × magnification, z = 10). For all cell counts, ten representative images were sampled from prospective motor (M1), somatosensory (S1) or visual (V1) primary cortices. After collapsing z-stacks, Lhx6-GFP positive neurons were manually counted using a 150 × 600 µm frame (i.e. $9 \times 10^4$ µm$^2$) using ImageJ 1.46j (National Institutes of Health, Bethesda, Maryland, USA). For quantification of cortical GABAergic interneurons in upper, middle and lower layers, the neocortex was divided in three frames, each of 150 × 600 µm in size, as indicated in *Figure 1—figure supplement 4*. Neonatal primary cortical areas were identified as described by Jacobowitz and Abbott (*Jacobowitz and Abbott, 1997*): primary motor cortex: P0/NB plate 5, primary somatosensory cortex: plate 5 to 6 and visual cortex: plate P0/NB plate 8 to 9. Parvalbumin and somatostatin cells were counted in each layer of postnatal day 21 neocortex in an area of 1280.35 µm$^2$ (10 images per region per mouse). Movement angle analysis was performed using ZEN measurement tool (ZEN Blue Edition, Zeiss, Jena, Germany). A straight line perpendicular to the cortical surface was drawn between the surface and the ventricle. The angle between this line and the movement vector of the cellular leading process was measured for each interneuron (identified by Lhx6-GFP expression) within a 150 × 600 µm frame. Morphological analysis of the leading process and classification according to morphological types I, IIa, IIb and III, was done according to Baudoin et al. (*Baudoin et al., 2012*).

## Statistical analysis

Statistical analyseis were made with Prism 5 (GraphPad Inc., La Jolla, CA, USA). Values in all graphs are shown as means ± standard error of the mean (SEM). An unpaired t-test was used for statistical evaluation. A p-value below 0.05 was considered as statistically significant.

## Acknowledgements

We thank James Li, Carmen Birchmeier, Kai Kaila, Jens Hjerling-Leffler, Juha Partanen, Thomas Hnasko and Ole Kiehn for providing mutant mice, Christoph Redies, Vassilis Pachnis and Alex Joyner for riboprobes, Hongbing Wang, Franck Polleux, Gord Fishell and Masanori Uchikawa for plasmids, Claudio Rivera for anti-KCC2 antibody, and Annika Andersson for technical assistance. Support for this research was provided by grants from the Swedish Research Council, Strategic Research Program in Regenerative Medicine, Knut and Alice Wallenbergs Foundation (Wallenberg Scholars Program), Karolinska Institute (Distinguished Professor Program), and National University of Singapore (to CFI) and Wenner-Gren Foundation (to SZ).

## Additional information

### Funding

| Funder | Author |
| --- | --- |
| Knut och Alice Wallenbergs Stiftelse | Carlos F Ibáñez |
| Karolinska Institutet | Carlos F Ibáñez |
| National University of Singapore | Carlos F Ibáñez |
| Wenner-Gren Foundation | Sabrina Zechel |
| Vetenskapsrådet | Carlos F Ibáñez |

The funders had no role in study design, data collection and interpretation, or the decision to submit the work for publication.

### Author contributions

SZ, Conception and design, Acquisition of data, Analysis and interpretation of data, Drafting or revising the article; YN, Analysis and interpretation of data, Contributed unpublished essential data or reagents; CFI, Conception and design, Analysis and interpretation of data, Drafting or revising the article

### Ethics

Animal experimentation: Animal protocols (N27/15; N173/15 and N240/13) were approved by Stockholms Norra Djurförsöksetiska nämnd and are in accordance with the ethical guidelines of the Karolinska Institute.

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
