## [Decision Letter]

Thank you for submitting your article "Thalamo-cortical axons regulate the radial dispersion of neocortical GABAergic interneurons" for consideration by *eLife*. Your article has been favorably evaluated by Marianne Bronner as the Senior Editor and three reviewers, one of whom is a member of our Board of Reviewing Editors. The reviewers have opted to remain anonymous.

The reviewers have discussed the reviews with one another and the Reviewing Editor has drafted this decision to help you prepare a revised submission.

Summary:

This interesting manuscript from Zechel and colleagues characterizes the role of thalamocortical axons (TCAs) in the radial migration of interneurons. It is known that interneurons colonize cortical layers in an inside-out pattern that is similar to pyramidal neurons, with early-born cells mainly populating deeper layers and late-born cells mainly committed to upper layers. Although some factors such as cues provided by pyramidal neurons and timing of neurogenesis influence the layering of interneurons, little is known about the mechanisms regulating this process. Here, the authors find that interneurons accumulate in lower cortical layers of mutant cortices lacking thalamocortical innervation. Interestingly, impaired radial migration of interneurons correlated with high levels of the KCC2 transporter whose removal was sufficient to restore interneuron lamination. Keeping KCC2 levels low, the authors argue, permits interneurons to migrate into the cortical plate. The authors go on to characterize the mechanisms controlling KCC2 levels and suggest that both disruption of interneuron NMDA receptors and inhibition of calpain activity lead to increased KCC2 expression.

This manuscript brings multiple transgenic mouse models to bear on this question. However, the observation of impaired interneuron migration in the absence of TCAs is more compelling than the proposed molecular mechanism and some additional data and discussion is required to support the overall story.

Essential revisions:

1) In Figure 4, there seems to be an increased proportion of interneurons expressing KCC2 in *Gbx2* KO cortices. However, the presented images are not very convincing representations of the quantification, and the data would be more compelling with better images.

2) The authors state that KCC2 upregulation impairs radial migration of interneurons because the defects observed in *Gbx2* KO cortices were rescued upon deletion of KCC2 in the double mutants (Figure 5). However, non-cell-autonomous effects cannot be ruled out on the basis that KCC2 was also depleted in cortical neurons, which affects their maturation and activity (Cancedda et al., 2007). To make the strong point that KCC2 expression in interneurons is causative for their migration would require cell-type specific manipulations of KCC2 expression selectively in interneurons, and although the reviewers would be very interested to see this, we do not consider this experiment required for this manuscript. However in lieu of those data, the authors need to back off on the mechanism and discuss the likelihood the effects are cell-autonomous or not.

3) In Figure 7 the authors show that depleting the NR2B NMDA subunit alters interneuron cortical layering. However, it is unclear whether these effects correlate with increased expression of KCC2 levels in interneurons, which would be expected on the basis of the model. The authors would strengthen their model (and provide an alternative way to deal with the concern in point 3) if they present images and quantify KCC2 protein levels in NR2B knockdown neurons similar to the quantification shown in Figure 4. We do not consider the experiment to be essential, but a discussion of this point should be added to the text if the experiment is not included.

4) It is a minimum standard in the field for knockdown experiments to show a second independent shRNA against the target of interest that shows the same phenotype. These experiments need to be included.

5) In Figure 9 the images were meant to demonstrate that there is a reduction of both parvalbumin and somatostatin interneurons in upper layers of the cortex. The effects are small and the images not very convincing. Moreover, the reduction in the numbers of interneurons populating upper layers was not accompanied by an accumulation in lower layers as shown in newborn sections. Could an increase in interneuron cell death account for the lack of accumulation in deeper layers and/or significant reduction in parvalbumin+ total numbers? These points should be discussed in the text.

---

## [Author Response]

*Essential revisions:*

*1) In Figure 4, there seems to be an increased proportion of interneurons expressing KCC2 in Gbx2 KO cortices. However, the presented images are not very convincing representations of the quantification, and the data would be more compelling with better images.*

Figure 4: We have included better images and higher magnifications as requested.

*2) The authors state that KCC2 upregulation impairs radial migration of interneurons because the defects observed in Gbx2 KO cortices were rescued upon deletion of KCC2 in the double mutants (Figure 5). However, non-cell-autonomous effects cannot be ruled out on the basis that KCC2 was also depleted in cortical neurons, which affects their maturation and activity (Cancedda et al., 2007). To make the strong point that KCC2 expression in interneurons is causative for their migration would require cell-type specific manipulations of KCC2 expression selectively in interneurons, and although the reviewers would be very interested to see this, we do not consider this experiment required for this manuscript. However in lieu of those data, the authors need to back off on the mechanism and discuss the likelihood the effects are cell-autonomous or not.*

Figure 5: We have included new data on KCC2 knockdown by in utero electroporation of anti-KCC2 shRNA in E13.5 wild type and *Gbx2* knockout embryos (Figure 5). Our electroporation method restricts delivery of the plasmids to the ventral telencephalon where the MGE and the GABAergic interneurons are initially located. We found that electroporation of KCC2 shRNA normalized the distribution of GABAergic interneurons in the *Gbx2* mutants, eliminating their differences compared to wild type controls. This result is in agreement with the data obtained in double knock-out embryos and supports a cell-autonomous mechanism. We therefore feel confident about our initial assessment.

*3) In Figure 7 the authors show that depleting the NR2B NMDA subunit alters interneuron cortical layering. However, it is unclear whether these effects correlate with increased expression of KCC2 levels in interneurons, which would be expected on the basis of the model. The authors would strengthen their model (and provide an alternative way to deal with the concern in point 3) if they present images and quantify KCC2 protein levels in NR2B knockdown neurons similar to the quantification shown in Figure 4. We do not consider the experiment to be essential, but a discussion of this point should be added to the text if the experiment is not included.*

We have added a new figure (Figure 7) showing elevated proportion of KCC2^+^ cells after electroporation of NR2B shRNAs compared to control shRNA. Thus, depleting the NR2B NMDA subunit alters interneuron cortical layering, and this correlates with increased expression of KCC2 levels in the interneurons. This result is in agreement with the negative regulation of KCC2 expression by NMDAR activity.

*4) It is a minimum standard in the field for knockdown experiments to show a second independent shRNA against the target of interest that shows the same phenotype. These experiments need to be included.*

Figure 7: We now show knockdown of NR2B expression by in utero electroporation using two independent shRNA sequences as requested.

*5) In Figure 9 the images were meant to demonstrate that there is a reduction of both parvalbumin and somatostatin interneurons in upper layers of the cortex. The effects are small and the images not very convincing. Moreover, the reduction in the numbers of interneurons populating upper layers was not accompanied by an accumulation in lower layers as shown in newborn sections. Could an increase in interneuron cell death account for the lack of accumulation in deeper layers and/or significant reduction in parvalbumin+ total numbers? These points should be discussed in the text.*

We have not detected increased cell death, as assessed by immunostaining for cleaved caspase-3, in neocortex of newborn or 3-week old *Olig3-Cre;Gbx2*fx/fx mice (data not shown), indicating that any cell loss should have occurred between P0 and P21. This is now discussed in the text as requested.